# Golgi apparatus-targeted aggregation-induced emission luminogens for effective cancer photodynamic therapy

Minglun Liu[1], Yuncong Chen [1,2 ✉], Yan Guo[1], Hao Yuan[1], Tongxiao Cui[1], Shankun Yao[1], Suxing Jin[1], Huanhuan Fan[1], Chengjun Wang[3], Ran Xie[1], Weijiang He [1,2 ✉] & Zijian Guo [1,2 ✉]

Golgi apparatus (GA) oxidative stress induced by in situ reactive oxygen species (ROS) could severely damage the morphology and function of GA, which may open up an avenue for effective photodynamic therapy (PDT). However, due to the lack of effective design strategy, photosensitizers (PSs) with specific GA targeting ability are in high demand and yet quite challenging. Herein, we report an aggregation-induced emission luminogen (AIEgen) based PS (TPE-PyT-CPS) that can effectively target the GA via caveolin/raft mediated endocytosis with a Pearson correlation coefficient up to 0.98. Additionally, the introduction of pyrene into TPE-PyT-CPS can reduce the energy gap between the lowest singlet state ($S_1$) and the lowest triplet state ($T_1$) ($\Delta E_{ST}$) and exhibits enhanced singlet oxygen generation capability. GA fragmentation and cleavage of GA proteins (p115/GM130) are observed upon light irradiation. Meanwhile, the apoptotic pathway is activated through a crosstalk between GA oxidative stress and mitochondria in HeLa cells. More importantly, GA targeting TPE-T-CPS show better PDT effect than its non-GA-targeting counterpart TPE-PyT-PS, even though they possess very close ROS generation rate. This work provides a strategy for the development of PSs with specific GA targeting ability, which is of great importance for precise and effective PDT.

[1] State Key Laboratory of Coordination Chemistry, School of Chemistry and Chemical Engineering, Chemistry and Biomedicine Innovation Center (ChemBIC), Nanjing University, Nanjing 210023, China. [2] Nanchuang (Jiangsu) Institute of Chemistry and Health, Nanjing 210000, China. [3] Sinopec Shengli Petroleum Engineering Limited Company, Dongying 257068, China. ✉email: chenyc@nju.edu.cn; heweij69@nju.edu.cn; zguo@nju.edu.cn

Photodynamic therapy (PDT) is an attractive tumor treatment, which could be spatially and temporally controlled by light[1]. It shows advantages such as minimal invasiveness, selective killing of tumor by light-induced cytotoxic reactive oxygen species (ROS), particularly singlet oxygen ($^1O_2$), and repeated administration without drug resistance, which is one of the major problems related to the use of chemotherapeutic drugs (e.g., cisplatin)[2]. However, traditional photosensitizers (PSs) tend to show diminished $^1O_2$ quantum yield at aggregated states due to the enhanced nonradiative decay rate, which might decrease their PDT efficiency[3–5]. In this regard, PSs with aggregation-induced emission (AIE) characteristics are superior to traditional PSs since they show both increased $^1O_2$ and fluorescence quantum yield upon aggregation[6,7]. In addition, AIE luminogens (AIEgens) with intramolecular charge transfer (ICT) effect could facilitate the intersystem crossing (ISC) by decreasing $\Delta E_{ST}$, which could be adopted for developing heavy-atom-free PSs that show minimum dark toxicity[8–10]. Thus, AIEgen based PSs could serve as appealing alternatives for traditional PSs and have attracted much attention for efficient PDT[11–14]

Studies have revealed that $^1O_2$ shows very short lifetime (half-life: 0.03 to 0.18 ms) and narrow diffusion distance in biological systems (a radius of $< 0.02\ \mu m$)[1]. Hence, it is desirable to develop PSs that could target organelles and generate high dosage of $^1O_2$ in situ to directly cause dysfunction of subcellular organelles and activate specific cell death signals. Several PSs targeting mitochondria[15–17], endoplasmic reticulum (ER)[18–20], lysosome[21,22] and cell membrane[23,24] have been reported and exhibited high PDT efficiency. However, due to the lack of effective Golgi apparatus (GA) targeting strategies, specific GA targeting AIEgen-based PSs have seldom been reported. Golgi apparatus is a central node at the intersection point between the exocytic and endocytic channels in intracellular membrane transporting, which plays a pivotal role in the classification of newly synthesized and recycled proteins and lipids to their final destinations[25]. Oxidative stress will induce significant damages on the structure and physiological function of GA[26]. These changes in GA may trigger and propagate downstream stress signals, leading to GA disruption or even apoptosis[27–30]. Therefore, GA targeting AIEgen based PSs could serve as good candidates for efficient PDT.

In this work, we report a series of AIEgen-based PSs that can target GA and their applications in efficient suppression of tumor cells by PDT induced GA oxidative stress (Fig. 1). Among the AIEgens, TPE-PyT-CPS shows the highest $^1O_2$ generation capacity, resulting in the decomposition rate of ABDA (a singlet oxygen indicator) up to 32.85 nmol per minute and excellent GA targeting ability with a Pearson's correlation coefficient of 0.98. Structure-property relationship studies indicate that the cyano-group induced molecular rod-like stacking plays a critical role for specific GA targeting, while the strong ICT process and pyrene group contributes for the fast ISC rate and high $^1O_2$ generation capacity. GA undergoes severe oxidative stress and fragmentation after ROS are generated in situ upon PDT. As shown in Fig. 1b, the structural protein p115 of GA is found to be cleaved into N-terminal and C-terminal fragments, the latter then translocates into the nucleus, resulting in upregulation of p53 and dysfunction of mitochondria and activation of apoptotic pathway. Finally, prominent inhibition of tumor cell growth is demonstrated without noticeable adverse effect. This study not only provides an efficient strategy for developing GA targeted PSs, but also offers important insights for efficient and precise treatment of cancers.

## Results

### Design, synthesis, spectroscopic and ROS generation properties.
Traditional PSs usually have to face problems such as aggregation caused quenching (ACQ) and potential dark toxicity due to heavy atom modification[31,32]. AIEgen based PSs with a heavy atom free D-π-A structure can efficiently separate the highest occupied molecular orbital (HOMO) and lowest unoccupied molecular orbital (LUMO) distribution, which is helpful to promote the rate of intersystem crossing ($k_{ISC}$) process and thus beneficial to increase $^1O_2$ generation[32]. TPE-PyT-CPS was designed with an AIE active tetraphenylethene (TPE) derivative as the electron-donating group (EDG) and a cyano-pyridinium salt moiety act as the electron-withdrawing group (EWG). To improve the ISC efficiency, a pyrene ring and thiophene group was introduced as π spacer for better separation of HOMO-LUMO and decreasing $\Delta E_{ST}$. The synthesis procedures of TPE-PyT-CPS and its control compounds (TPE-PyT-CP, TPE-PyT-PS, TPE-T-CPS) were presented in Supporting Information. All the intermediates and target compounds were fully characterized by $^1H$ and $^{13}C$ NMR spectroscopy and HRMS.

The spectroscopic properties of all compounds in water were studied with the UV − vis and photoluminescence (PL) spectrometer (Supplementary Fig. 18), in which the TPE-PyT-CPS shows a maximum absorption peak at 487 nm and a Near-infrared (NIR) emission at 680 nm in aqueous solution, respectively. The Stokes shift was nearly 200 nm, which can reduce the interference of background signal and helps to improve the image-guided therapy. Moreover, compared with the TPE-PyT-PS and TPE-PyT-CP, the UV absorption spectra and fluorescence emission spectra of TPE-PyT-CPS and TPE-T-CPS show a significant redshift, which is due to the stronger ICT effect caused by EDGs of cyano-group and pyridinium salt group. On the other hand, those AIEgens exhibited good photostability compared with the ICG (Indocyanine green) under laser irradiation for 30 min (Supplementary Fig. 19, Supplementary Table 3). As depicted in Fig. 2a and b TPE-PyT-CPS in pure acetonitrile exhibited very weak emission, while with the water fraction increasing to 30%, the emission enhanced slightly. Further increasing the water fraction to 99%, lead to a distinct enhancement on PL intensity with a gradual redshift of the emission maximum to 680 nm, indicating typical AIE characteristics of TPE-PyT-CPS. The fluorescence quantum yields of TPE-PyT-CPS in acetonitrile and water (with 1% acetonitrile, $v/v$) were 2.74% and 17.1%, respectively. In addition, the AIE features of the other three AIEgens were also confirmed using a mixed solvent (acetonitrile and water) system with different water fractions (Supplementary Fig. 20, Supplementary Table 1).

Next, transmission electron microscopy (TEM) and dynamic light scattering (DLS) assays were conducted to characterize the morphology and particle size of the AIEgens in aqueous solution (Supplementary Fig. 21). The results showed that the particle size of TPE-PyT-CPS in water and acetonitrile were around 410 nm and 1 nm, respectively (Fig. 2c). The results suggested that the TPE-PyT-CPS is almost monodisperse in acetonitrile but form aggregates (rod-like according to TEM) in aqueous solution and the phenomenon of rod-like stacking can also be observed by DMEM (Supplementary Fig. 22). The in vitro $^1O_2$ generation capability of TPE-PyT-CPS was evaluated under 532 nm laser irradiation with 9,10-anthracenediyl-bis(methylene)dimalonic acid (ABDA) as the $^1O_2$ indicator (Fig. 2d-f). Upon light irradiation, the absorbance of ABDA in acetonitrile showed almost no change within 12 min in the presence of TPE- PyT-CPS, suggesting poor $^1O_2$ generation capability even though acetonitrile can dissolve more oxygen[33–35]. However, the absorbance of ABDA in water (contains 1% acetonitrile) decreased sharply upon light irradiation for 120 sec, indicating that TPE-PyT-CPS possessed a dramatically elevated ROS in the aggregation state. Interestingly, we found that the TPE-PyT-PS that without -CN group can form smaller particles (7.2 nm) than

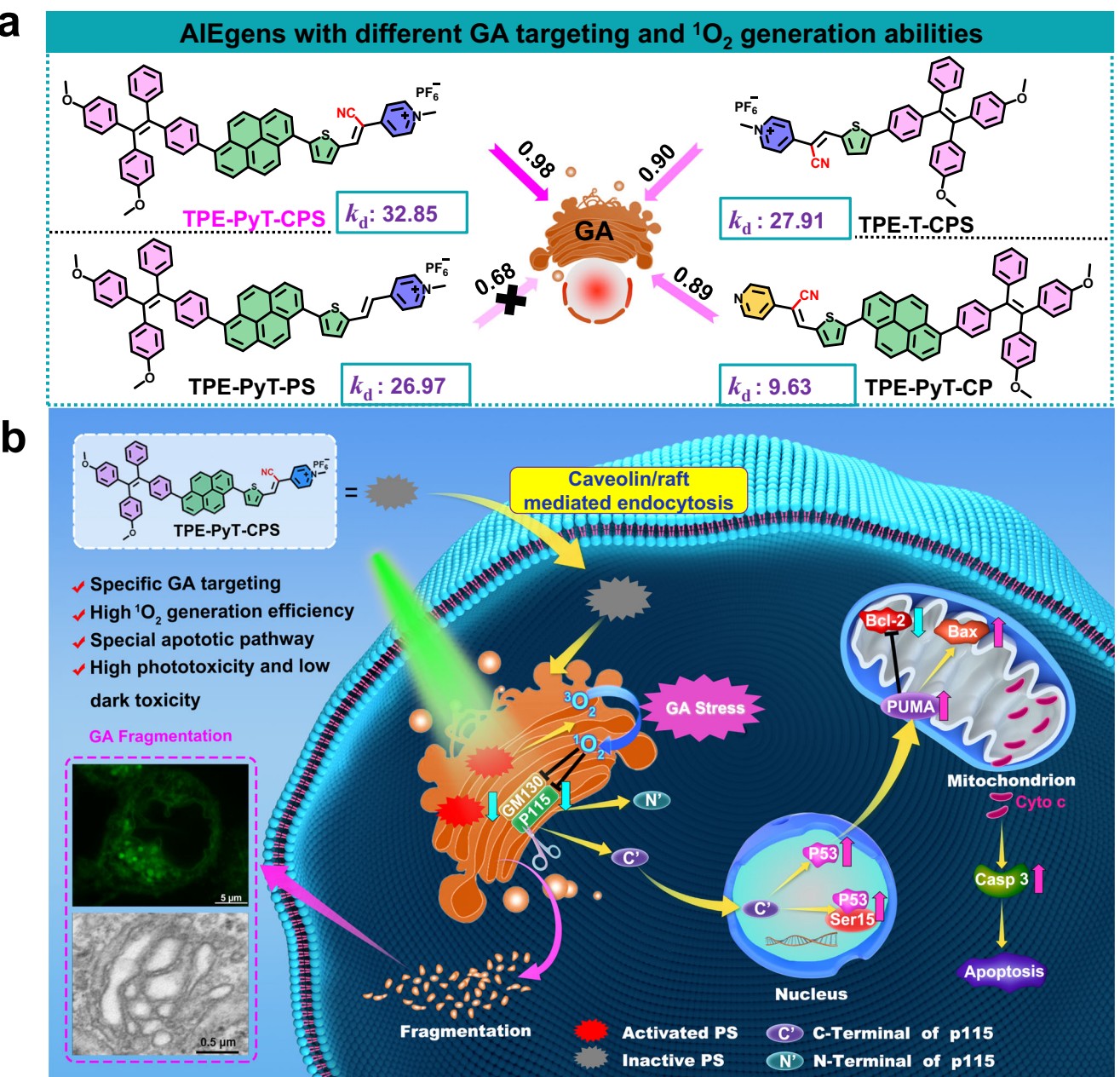

**Fig. 1 GA targeting and ROS generation capacity of the AIEgens, as well as the apoptosis pathway caused by Golgi oxidative stress. a** The chemical structures of AIEgens with different GA targeting (PCC values) and ROS generation abilities ($k_d$: nmol/min): decomposition rates of ABDA in the presence of different AIEgens. **b** Schematic illustration of AIEgen induced GA stress and the crosstalk between GA and mitochondria for cell apoptosis upon PDT.

the rod-like morphology of other AIEgens (200-400 nm) in aqueous solution, which may due to the highly polar cyano group can improve the order of molecular arrangement by affecting the molecular orientation[36–38].

In addition, the fluorescence of HPF (superoxide anion probe) and DHR123 (hydroxyl radical probe) did not change significantly in the presence of TPE-PyT-CPS upon laser irradiation, indicating that TPE-PyT-CPS generates ROS mainly through type II PDT process (Supplementary Fig. 24). Through the $^1O_2$ generated by TPE-PyT-CPS, we obtained the decomposition rate ($k_d$) on ABDA in aqueous solution was 32.85 nmol per minute, while those of the control compounds were found to be 27.91, 26.97 and 9.63 nmol per minute for TPE-T-CPS, TPE-PyT-PS and TPE-PyT-CP, respectively (Supplementary Fig. 23)[39]. TPE-PyT-PS showed much higher $^1O_2$ generation capability

than TPE-PyT-CP, suggesting that the pyridinium salt group contributed more for ROS generation than the –CN group. The stronger electron-withdrawing ability of the pyridinium salt group than –CN group leads to stronger ICT effect and more efficient ISC of TPE-PyT-PS than that of TPE-PyT-CP. It is reasonable that TPE-PyT-CP and TPE-PyT-PS showed lower ROS generation yields than TPE-PyT-CPS, since the former two showed weaker ICT effects due to the lack of EDGs of either the pyridinium group or –CN group.

TPE-PyT-CPS and TPE-T-CPS showed identical electron donor and acceptor but different π spacers and $^1O_2$ generation ability. To understand the mechanism behind this phenomenon, density functional theory (DFT) calculations were conducted. As shown in Fig. 3a, although the HOMO and LUMO electronic distributions of TPE-T-CPS are well separated, there are some overlaps at the region

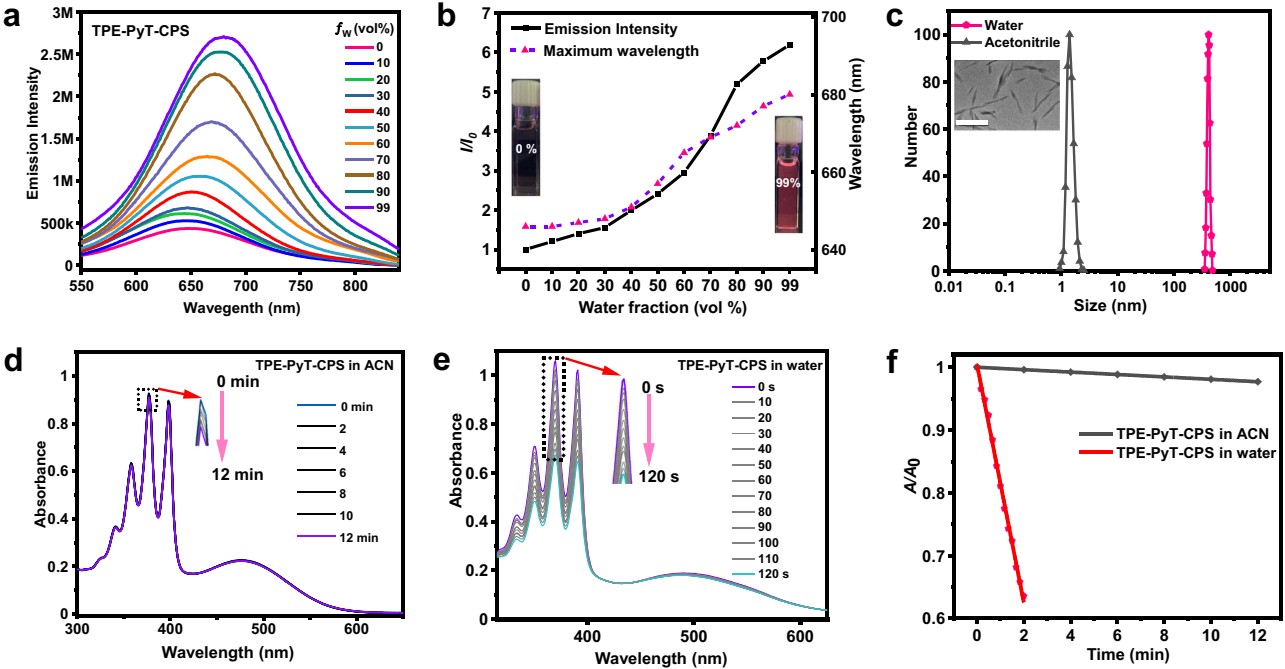

**Fig. 2 AIE characteristics of TPE-PyT-CPS and its ROS generation ability in different solvents. a** Emission spectrum of TPE-PyT-CPS in acetonitrile/water mixtures with varied water fractions ($f_w$). **b** Emission intensity plot of $I/I_0$ (dark dot) and the maximum emission wavelength (red dot) of TPE-PyT-CPS in different $f_w$ of the solvent mixture, $I_0$ represents the emission intensity in acetonitrile. **c** The particle sizes of TPE-PyT-CPS (10 μM) in water and acetonitrile was measured by dynamic light scattering. Inset: TEM images of TPE-PyT-CPS in water. Scale bar:1μm. Changes of UV-vis spectra of ABDA (60 μM) in the presence of TPE-PyT-CPS (10 μM) in acetonitrile (**d**) and water (**e**) under different durations of light irradiation (10 mW cm$^{-2}$). **f** The decomposition rates of ABDA in different solvents in the presence of TPE-PyT-CPS under light irradiation ($A_0$ and A represent the absorption of ABDA before and after laser irradiation, respectively). Experiments in 2a-f were performed two times independently and experiments in 2c (insert) were performed three times independently, representative images are shown. Source data are provided as a Source Data file.

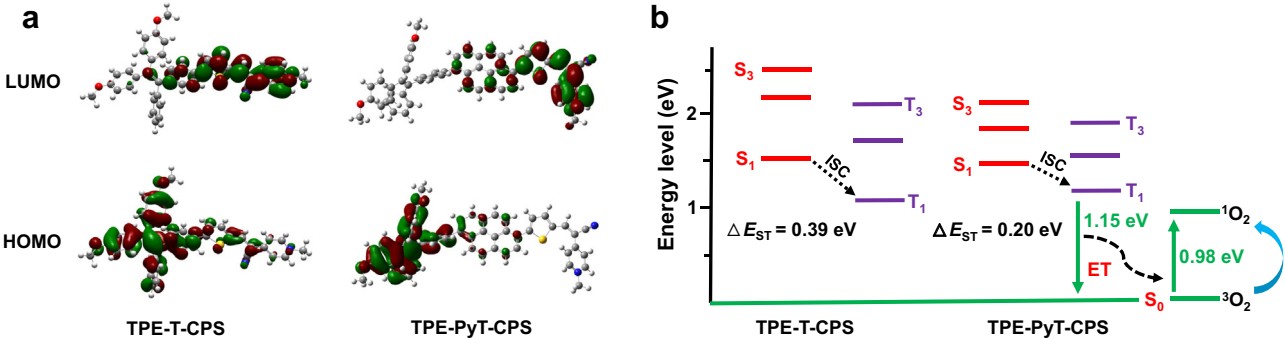

**Fig. 3 Calculated frontier molecular orbitals and energy levels for TPE-T-CPS and TPE-PyT-CPS. a** HOMO–LUMO electronic cloud distributions of TPE-T-CPS and TPE-PyT-CPS by DFT calculations. **b** Energy levels ($S_1$–$S_3$, $T_1$–$T_3$) of TPE-T-CPS and TPE-PyT-CPS calculated by the vertical excitation of the optimized structures. ET: energy transfer.

of thiophene and the nearby phenyl ring of TPE-T-CPS. However, by introducing an additional large π spacer pyrene, the overlap between HOMO and LUMO of TPE-PyT-CPS was distinctly reduced. As a result, the $\Delta E_{ST}$ value of TPE-PyT-CPS was calculated to be 0.20 eV, smaller than that of TPE-T-CPS, which was calculated to be 0.39 eV (Fig. 3b, Supplementary Table 2). The DFT calculation was consistent with the experimental data that TPE-PyT-CPS showed higher $^1O_2$ generation capacity than TPE-T-CPS in water, which verified our design strategy that the introduction of pyrene spacer can help improve the HOMO-LUMO separation within the ICT system, thus elevate the $^1O_2$ generation ability.

**Subcellular localization study.** The intracellular localization of TPE-PyT-CPS was investigated by confocal laser scanning microscopy (CLSM). As shown in Fig. 4a, the PCC between fluorescence images of marked commercial probes and the TPE-PyT-CPS were determined to be 0.36 for Mito, 0.54 for ER, 0.98 for GA and 0.61 for lysosome, respectively. Moreover, we evaluated the distribution of TPE-PyT-CPS in different organelles with the organelle separation kits combined with UV-vis measurement. As shown in Fig. 4b, we found the TPE-PyT-CPS prominently accumulation in the Golgi apparatus, and the absorption spectra indicated that the distribution of TPE-PyT-CPS in the Golgi apparatus was at least 8 times higher than that of other major organelles (Fig. 4c, Supplementary Fig. 28). The results clearly demonstrated that TPE-PyT-CPS localized preferentially in GA with high specificity. In addition, the PCC values between the fluorescence images of GA marker and those

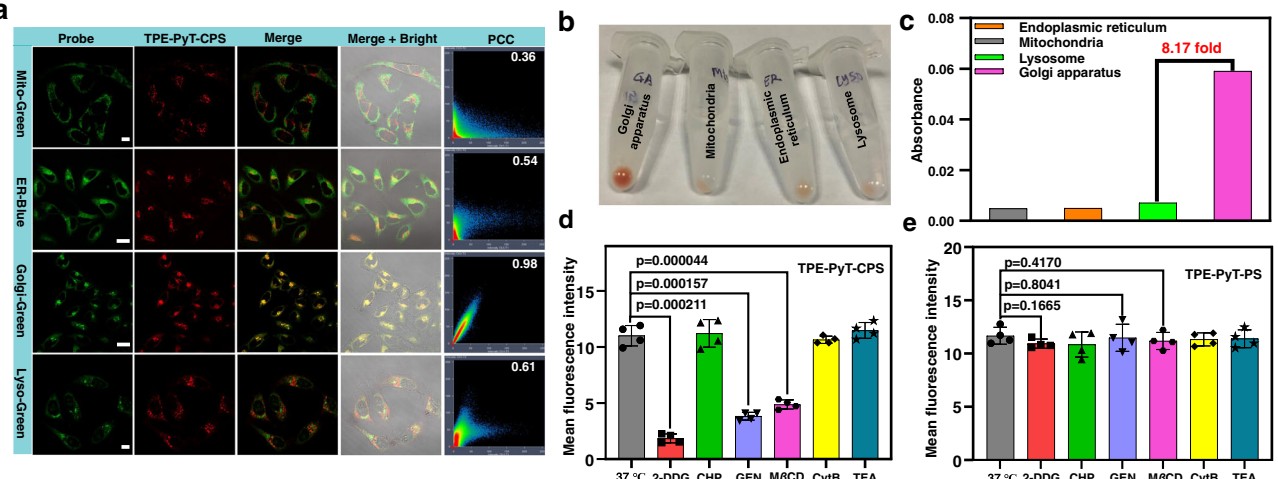

**Fig. 4 Subcellular co-localization of TPE-PyT-CPS and the uptake behavior difference between TPE-PyT-PS and TPE-PyT-CPS. a** CLSM of HeLa cells stained with different commercial probes (Mito-Green: 495-535 nm, $\lambda_{ex}$, 488 nm. ER-Blue: 455-520 nm, $\lambda_{ex}$, 405 nm. Golgi-Green: 495-535 nm, $\lambda_{ex}$, 488 nm. Lyso-Green: 495-535 nm, $\lambda_{ex}$, 488 nm) and TPE-PyT-CPS (10 µM, 600-750 nm, $\lambda_{ex}$, 488 nm). Scale bar: 20 µm for ER and Golgi channel;10 µm for Mito and Lyso channel. **b** Distribution images and UV absorption of TPE-PyT-CPS (c) in each organelle after separation by subcellular organelle kits. Average fluorescence intensity of HeLa cells after incubated with TPE-PyT-CPS (**d**) and TPE-PyT-PS (**e**) in the presence of different endocytosis inhibitors including 2-DDG, CHP, GEN, MβCD, CytB and TEA. Experiments in 4a were performed three times independently and experiments in 4b-c were performed two times independently, representative images are shown. Data in d and e are presented as mean ± SD derived from $n = 4$ biologically independent experiments. Statistically significant differences between the experimental groups were analyzed by two-tailed Student's t-test. when $p < 0.05$, it was considered to have statistical significance. Source data are provided as a Source Data file.

of each compound were determined to be 0.98, 0.90, 0.89 and 0.68 for TPE-PyT-CPS, TPE-T-CPS, TPE-PyT-CP and TPE-PyT-PS, respectively (Fig. 1a, Supplementary Fig. 25-27). Only TPE-PyT-PS without the –CN group showed poor GA targeting ability while the other three showed specific GA distribution.

Next, we try to elucidate the subcellular distribution mechanism of different AIEgens. Firstly, we tested the lipophilicity of all compounds (Supplementary Fig. 29, Supplementary Table 2) because of lipophilicity was reported to play an important role in the cell uptake and distribution of compounds in subcellular organelles[40]. The log $P_{O/W}$ values of each compound were determined to be 1.764, 1.581, 2.694 and 2.201 for TPE-PyT-CPS, TPE-T-CPS, TPE-PyT-CP and TPE-PyT-PS, respectively. The log $P_{O/W}$ value of these AIEgens obeys the QSAR model, which claimed that probes with GA targeting ability show log $P$ values between 0-8[41,42]. Next, we studied the cell uptake pathways of AIEgens using different biochemical inhibitors. The results (Fig. 4d, e and Supplementary Fig. 30, 31) demonstrated that there were no distinct differences in cell uptake for TPE-PyT-PS after treated with all of the blockers. Moreover, the internalization rate of TPE-PyT-PS was close to saturation after incubated with HeLa cells for 3 h, while the other three AIEgens reached saturation after 6 h (Supplementary Fig. 32, 33). The smaller particle size (7.2 nm) of TPE-PyT-PS aggregates showed a faster internalization rate in an energy-independent manner, which was distinctly different from the other three AIEgens. The cell uptake of TPE-PyT-CPS significantly decreased by 82.9% upon the inhibition of 2-DDG (an ATP synthesis inhibitor), suggesting the energy-dependent uptake manner. Moreover, the cell uptake of TPE-PyT-CPS showed a decrease of 65.8% and 55.5% in the presence of GEN (a caveolae-mediated endocytosis inhibitor) and MβCD (a lipid raft mediated endocytosis inhibitor), respectively, indicating a caveolin/raft mediated endocytosis pathway. Similar endocytosis patterns were found for TPE-T-CPS and TPE-PyT-CP. The results were in accordance with the reports which claimed that small molecules could target GA via the caveolin/raft mediated endocytic pathway [43–45].

**ROS generation and PDT in tumor cells.** The high $^1O_2$ generation ability and excellent GA targeting ability of TPE-PyT-CPS encouraged us to investigate its ability to kill cancer cells through PDT. The in situ $^1O_2$ generation capability was evaluated by detecting the fluorescence intensity of Singlet Oxygen Sensor Green (SOSG). As shown in Fig. 5a, TPE-PyT-CPS, TPE-T-CPS and TPE-PyT-PS treated HeLa cells showed strong green fluorescence under laser irradiation, while almost no fluorescence was observed under dark condition. Under the same condition, SOSG signal of cells that treated with TPE-PyT-CPS was higher than that of TPE-T-CPS and TPE-T-PS, while the latter two has almost equal ROS generation capacity (Fig. 5b). This was in good accordance with the fact that TPE-PyT-CPS has higher singlet oxygen generation capacity. Subsequently, the PDT effects of TPE-PyT-CPS, TPE-T-CPS and TPE-PyT-PS were evaluated by 3-(4,5-dimethy lthiazol-2-yl)-2,5-diphenyltetra-zolium bromide (MTT) assay. Upon 532 nm laser irradiation (65 mW cm$^{-2}$) for 2 min, dose-dependent toxicities were observed and half-maximal inhibitory concentrations (IC$_{50}$) to HeLa cells were determined to be 170 nM, 400 nM and 1331 nM for TPE-PyT-CPS, TPE-T-CPS and TPE-PyT-PS, respectively (Fig. 5c). However, negligible dark toxicities were observed upon incubation of HeLa cells with TPE-PyT-CPS, TPE-T-CPS and TPE-PyT-PS (up to 256 µM) for 24 h (Supplementary Fig. 34), demonstrating exceptionally high phototoxicity index (PI = IC$_{50}$ dark/IC$_{50}$ light) values over 1500, 640, and 192, respectively. The above results show that TPE-PyT-CPS has significant ROS generation ability and the best inhibitory effect on the growth of tumor cells. In particular, we found that under the condition of nearly equal ROS production capacity, TPE-T-CPS showed better cytotoxicity than TPE-PyT-PS, indicating that targeting Golgi is indeed conducive to improving the effect of photodynamic therapy. TPE-PyT-CPS was then selected for the following mechanistic and in vivo PDT studies, since it showed brilliant PDT performance in vitro.

To study the cell death mechanism of tumor cells, Annexin V-FITC and propidium iodide (PI) were used to monitor the process of cell apoptosis induced by TPE-PyT-CPS mediated

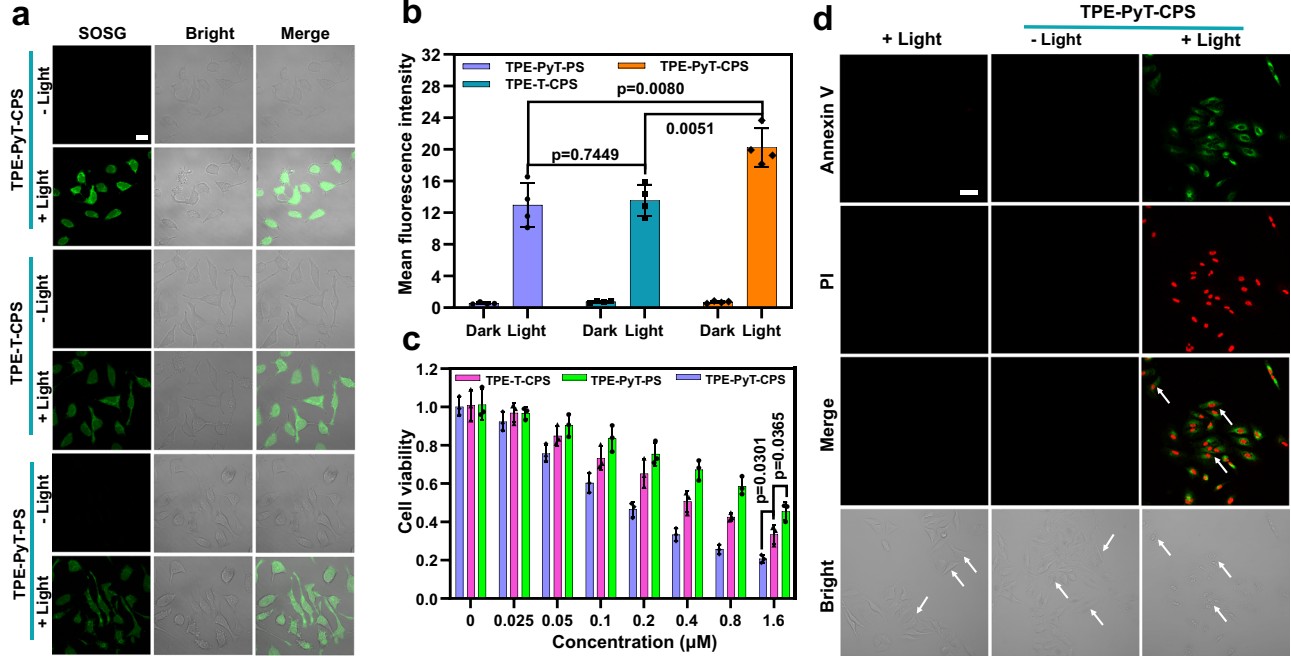

**Fig. 5 Intracellular ROS, cytotoxicity and cells apoptosis induced by PDT. a** Intracellular $^1O_2$ detection by CLSM after HeLa cells were incubated with different AIEgens including TPE-PyT-CPS (0.2 μM), TPE-T-CPS (0.4 μM), TPE-PyT-PS (0.4 μM) and SOSG under laser irradiations (25 mW cm$^{-2}$, 2 min). Scale bar, 20 μm. **b** Average fluorescence intensity of SOSG after the HeLa cells incubated with different AIEgens and under laser irradiations for 2 min (25 mW cm$^{-2}$, n = 4 biologically independent experiments). **c** Cell viabilities of HeLa cells after incubation with varied concentrations of TPE-T-CPS, TPE-PyT-PS and TPE-PyT-CPS with 532 nm laser irradiation for 2 min (65 mW cm$^{-2}$, n = 3 biologically independent experiments). **d** Apoptosis of HeLa cells treated with TPE-PyT-CPS (0.2 μM) under 532 nm laser (65 mW cm$^{-2}$, 2 min). Scale bar, 50 μm. Experiments in 5a and 5d were performed three times independently, representative images are shown. Data in 5b and 5c are presented as mean ± SD. Statistically significant differences between the experimental groups were analyzed by two-tailed Student's t-test. when $p < 0.05$, it was considered to have statistical significance. Source data are provided as a Source Data file.

PDT. HeLa cells were incubated with TPE-PyT-CPS and irradiating with 532 nm laser (65 mW cm$^{-2}$, 2 min), followed by Annexin V-FITC and PI staining, then the fluorescence images were collected using CLSM. As shown in Fig. 5d, the green and red fluorescence signals were hardly detected with irradiation alone or with TPE-PyT-CPS but without irradiation. However, after TPE-PyT-CPS incubation and light irradiation for 2 min, distinct green and red fluorescence were observed in HeLa cells, suggesting the occurrence of cell apoptosis. In addition, compared with the control group, the cells treated with TPE-PyT-CPS and light irradiation, showed significant shrinkage as indicated by the white arrow in the bright fields. Moreover, flow cytometry analysis demonstrated that the apoptosis rate of HeLa cells reached 56.7% when incubated with TPE-PyT-CPS (0.2 μM) under light irradiation (Supplementary Fig. 35). The results indicated that apoptosis was the main death pathway of tumor cells upon PDT treatment.

**PDT induced GA morphology change**. TPE-PyT-CPS was confirmed to target GA specifically and generate abundant ROS in situ, which could induce severe damage on structural proteins of GA and cause morphological and functional changes of GA[46,47]. Therefore, we used CLSM and TEM to monitor the ultrastructural alterations of GA in HeLa cells caused by TPE-PyT-CPS (Fig. 6). When the cells were treated with light only or with TPE-PyT-CPS (0.2 μM) in the dark, negligible effect on the GA morphology was observed. However, when TPE-PyT-CPS (0.2 μM) and light were present (i.e., PDT) in unison, the Golgi structure of HeLa cells changed obviously, including intensive swelling and fragmentation of GA cistern. The results confirmed

that TPE-PyT-CPS could cause severe damage on GA structure by in situ $^1O_2$ generation during PDT.

**GA oxidative stress caused apoptosis pathway**. The morphological changes of Golgi apparatus can activate the related signal pathways and trigger cell repair or apoptosis[48]. p115 is a vesicle tethering protein, which functions in Golgi-vesicle tethering and Golgi-cisternal stacking by bridging GM130 and giantin[49]. To further ascertain whether these structural proteins were affected during GA-targeted PDT process, the expressions of p115 and GM130 in HeLa cells were evaluated by western blot. As shown in Fig. 7a and b, the expression of GM130 and p115 in HeLa cells was gradually decreased with the extension of irradiation time during PDT, especially for GM130. In addition, we found two fragments of 30 kD (C-terminal fragment) and 90 kD (N-terminal fragment) proteins which originated from p115 during this process, confirming that p115 was cleaved after GA oxidative stress. Moreover, the western blot experiments demonstrated that p53 and its phosphorylation derivative (Ser15) were significantly upregulated, which suggested that PDT induced GA stress was related to apoptosis.

Studies have suggested that the 30 kD residue of p115 can translocate into the nucleus through sumoylation and then interact with p53, leading to p53 phosphorylation, which leads to apoptosis pathway[50,51]. To verify this mechanism, we detected the level of p53 upregulated modulator of apoptosis (PUMA), which was located in the outer membrane of the mitochondria[52]. Indeed, the dramatically increased expression of pro-apoptotic protein PUMA and Bax was observed, accompanied with a downregulation of anti-apoptotic Bcl-2 in HeLa cells during PDT, resulting an increased Bax/Bcl-2 ratio (Fig. 7c, d). In addition, an

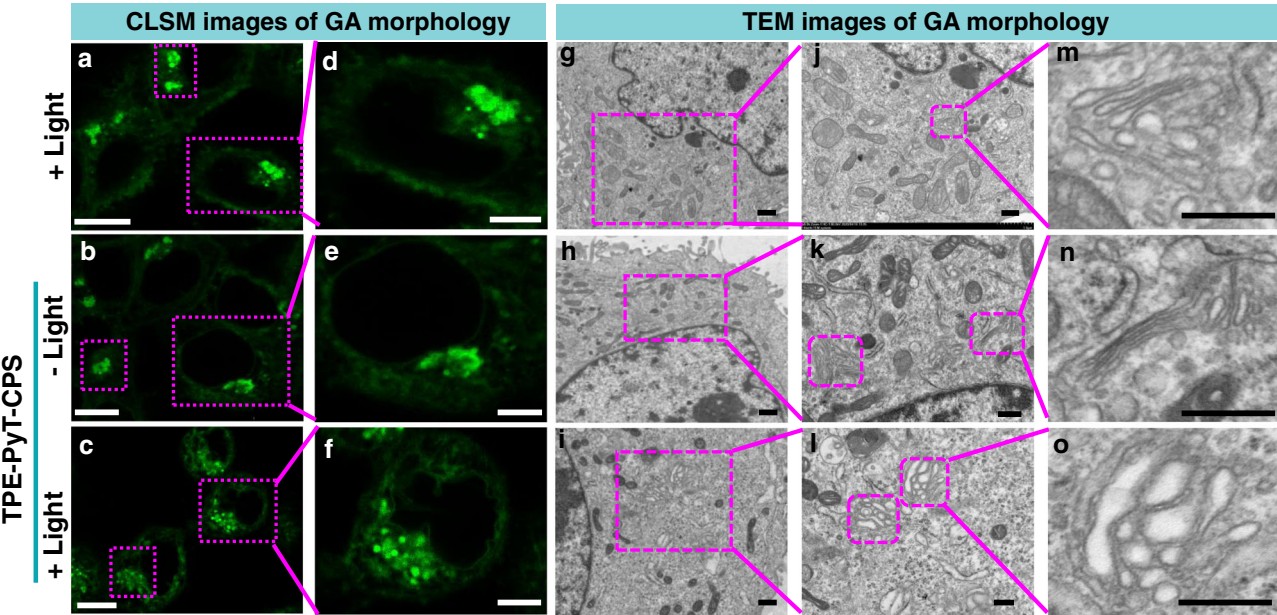

**Fig. 6 Morphological changes of Golgi apparatus after oxidative stress upon PDT treatment.** Morphological changes of GA observed by (**a–f**) CLSM (stained with Gogli Green) and TEM (**g–o**) imaging in HeLa cells after treatment with light only, TPE-PyT-CPS (0.2 μM) with or without light irradiation (25 mW cm$^{-2}$, 2 min). The experiments were performed three times independently, representative images are shown. Scale bars for **a–c**: 10 μm; **d–f**: 5 μm; **g–i**: 2 μm; **j–l**: 1 μm; **m–o**: 0.5 μm.

obvious upregulation of cleaved caspase-3 (a mitochondria-apoptotic protein) was also observed.

The activation of Bax can cause its conformational change and then oligomerization in the outer membrane of mitochondria, which will reduce the mitochondrial membrane potential (MMP, $\Delta\Psi_m$)[53,54]. Next, the MMP was evaluated through JC-1 staining assay. As shown in Fig. 8, the HeLa cells after PDT treatment showed an evident increase in green fluorescence of JC-1 monomers and a significant decrease in red channel of JC-1 aggregates, suggesting a distinct loss of MMP upon PDT. However, HeLa cells treated by light only, dark and "dark + AIEgen" showed intense red fluorescence and weak green fluorescence, indicating an intact MMP. The above results suggested that PDT induced GA oxidative stress could severely affect mitochondria homeostasis and active the apoptosis signal pathway (Fig. 9).

**GA-targeting mediated PDT in vivo.** The in vivo PDT performance of TPE-PyT-CPS was evaluated in a subcutaneous HeLa tumor-bearing mice model. All animal experiments were carried out in accordance with the regulations of the Institutional Animal Care and Use Committee (IACUC). Firstly, we evaluated the imaging ability of TPE-PyT-CPS in mice (Fig. 10a) and monitored the fluorescence signals at specified intervals. Distinct fluorescence signal in the tumor sites was observed after intratumoral injection of TPE-PyT-CPS (0.1 mM, 120 μL/200 mm$^3$ tumor) for 3 h (Supplementary Fig. 36), and the fluorescence intensity reached maximum after 18 h. To gain the fluorescence distribution image of the tumor and other major organs, the mice were sacrificed at 30 h after injection of TPE-PyT-CPS, and the ex vivo fluorescence images of isolated organs were shown in Supplementary Fig. 36. The results showed that TPE-PyT-CPS could effectively retain in the tumor tissue and exhibit strong fluorescence signal. Moreover, there was almost no fluorescence signals originated from other major organs, including heart, spleen, lung and kidney, except for a weak fluorescence

in liver tissue. The results demonstrated that TPE-PyT-CPS could retain in tumor area and serve as an imaging agent.

Subsequently, the anticancer efficacy of the AIEgens were assessed in vivo. Tumor (HeLa) bearing Balb/c-*nu* female mice were randomly divided into seven groups ($n = 5$): saline with laser irradiation group (saline + L), AIEgens groups that without laser irradiation (TPE-PyT-PS, TPE-T-CPS, TPE-PyT-CPS) and AIEgens with laser irradiation group (TPE-PyT-PS + L, TPE-T-CPS + L, TPE-PyT-CPS + L). Then the photosensitizers (0.2 mM, 100 μL in saline) were injected into mice in each administration group by intratumoral injection, in which the volume of mice tumor at the time of treatment was 119.05 ± 5.80 mm$^3$. As can be seen from Fig. 10b, for the control groups (saline + L, TPE-PyT-PS, TPE-T-CPS, TPE-PyT-CPS), the tumor volumes increased rapidly, suggesting that the tumor growth was not affected in the absence of AIEgens or only in light irradiation. However, in the presence of our AIEgens, the tumor volumes and weight were effectively suppressed in the light irradiation groups (TPE-PyT-PS + L, TPE-T-CPS + L, TPE-PyT-CPS + L) (Fig. 10b–d). Specifically, we found that TPE-PyT-CPS was more efficient than TPE-T-CPS ($p < 0.01$) in tumor growth inhibition, confirming that introducing pyrene in the molecular skeleton was helpful for PDT. More importantly, TPE-T-CPS that with GA targeting ability showed better therapeutic effect than TPE-PyT-PS ($p < 0.05$) that without GA targeting ability in terms of both tumor volume (Fig. 10c) and tumor weight (Fig. 10d), even though they possess very similar ROS generation capacity, indicating that GA targeting photosensitizers could be beneficial for tumor growth inhibition. Additionally, the body weights of mice in different treating groups (Fig. 10e) did not show significant changes. Hematoxylin and eosin (H&E) staining in tumors tissues of mice analysis confirmed that "AIEgens + L" groups exhibited obviously pyknotic cells with condensed nuclei, which was considered to be dead or apoptotic cells, whereas the of mice that treated in control groups had massive viable cancer cells (Fig. 10f, Supplementary Fig. 37)[55]. In addition, the terminal deoxynucleotidyl transferase-mediated dUTP-biotin nick end

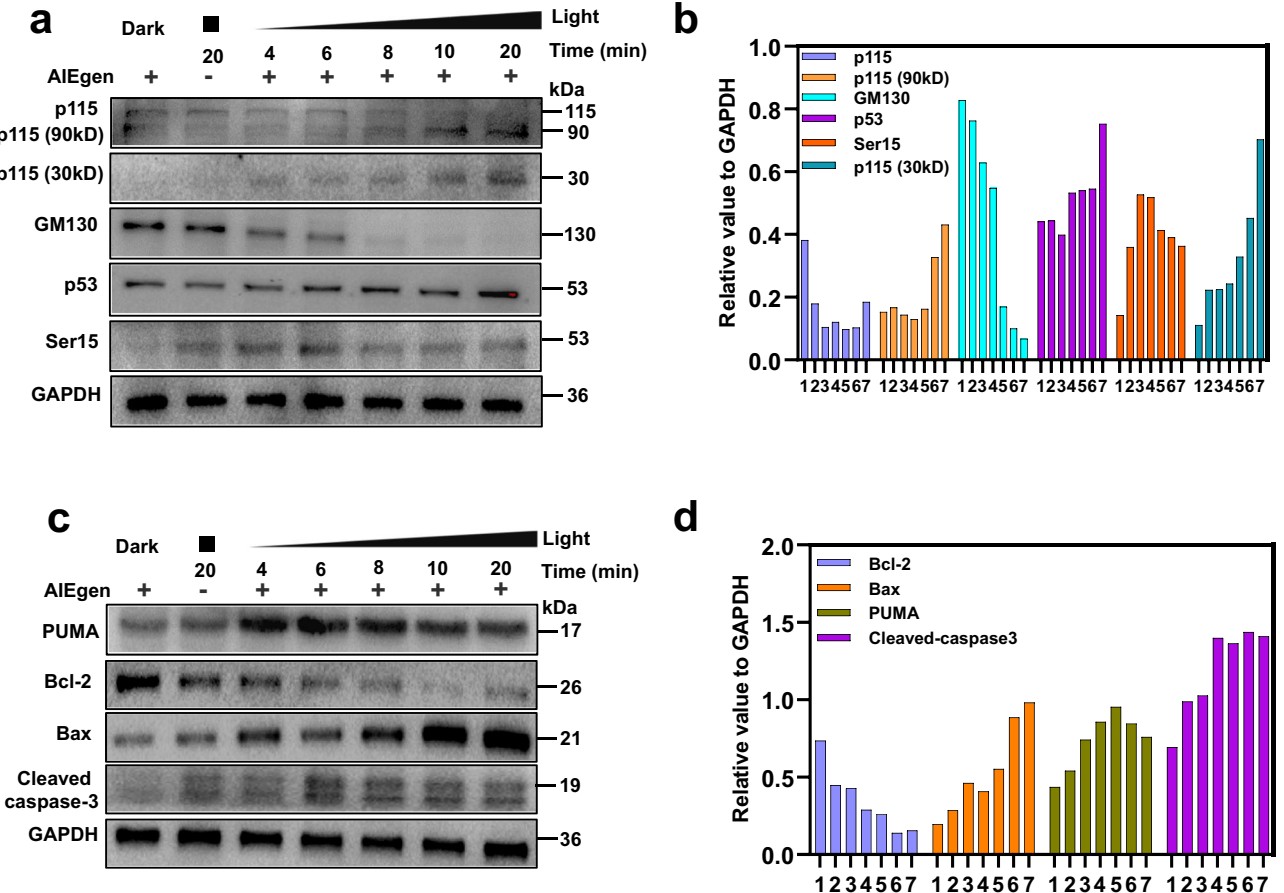

**Fig. 7 Golgi apparatus and mitochondria associated protein expression after PDT.** The expressions of p115, p115 (90 kD), p115 (30kD), GM130, p53, ser15 in Golgi apparatus (**a**) and PUMA, Bcl-2, Bax, caspase-3 in mitochondria (**c**) after different treatment with TPE-PyT-CPS (0.2 μM) under light irradiation (25 mW cm$^{-2}$). (**b**) and (**d**) western blot statistical chart of the relative expression levels of protein expressions in different treating groups (1. dark; 2. only light, 20 min; 3-7: with light and TPE-PyT-CPS for 4, 6, 8, 10 and 20 min, respectively) by imageJ and the value of protein expression is relative to GADPH. The experiments were performed two times independently with similar results, representative images are shown. Source data are provided as a Source Data file.

labeling (TUNEL) was adopted to study the antitumor mechanism of those AIEgens in mice. As illustrated in Fig. 10f and Supplementary Fig. 36, the AIEgens mediated PDT is found to exhibited significantly higher level of TUNEL-positive apoptotic cells that marked with green fluorescence than those of control groups.

Furthermore, the in vivo toxicology of the AIEgens was investigated by testing the blood biochemical index and the blood routine of Balb/c-*nu* female mice to evaluate the safety of photosensitizers in PDT. As shown in Supplementary Fig. 38, the blood biochemical index with various parameters including aspartate transaminase (AST), alanine transaminase (ALT), blood urea nitrogen (BUN), creatinine (CREA), creatinine and alkaline phosphatase (GGT) were detected, in which the AST, ALT and CREA were related to liver and kidney function of mice[56]. The results demonstrated that there were no significant differences ($p > 0.05$) between the saline group and the AIEgens groups, regardless of whether the AIEgens was irradiation or not. On the other hand, the haematology markers that consists of white blood cells (WBC), red blood cells (RBC), mean corpuscular volume (MCV), haemoglobin (HGB), mean corpuscular haemoglobin (MCH) and platelets (PLT) were measured. All the parameters of those markers between the saline group and AIEgens treated groups exhibited no obvious differences ($p > 0.05$). Meanwhile, there was no abnormal pathological morphology or prominent tissue damage in the heart, liver, spleen, lung and kidney of mice

in all treatment groups from the H&E staining (Supplementary Fig. 39). Taken together, these results confirmed that TPE-PyT-CPS could efficiently suppress tumor growth by GA targeting mediated-PDT with negligible side effects.

## Discussion

In conclusion, we have designed and synthesized a series of AIEgen based PSs with excellent GA targeting abilities and PDT effects. The cell uptake mechanism demonstrated that caveolin/raft mediated endocytosis of TPE-PyT-CPS was responsible for GA targeting, while the introduction of large π spacer pyrene group further enhanced the singlet oxygen generation ability of the D-π-A system. Distinct morphological change of GA was observed upon the in situ generation of ROS during PDT. Moreover, we found the GA stress can trigger the mitochondria dysfunction during PDT. The GA-mitochondria crosstalk led to the collapse of MMP and ultimately cell apoptosis. TPE-PyT-CPS showed excellent phototoxicity and negligible dark toxicity to HeLa cancer cells with a large PI over 1500. Meanwhile, benefiting from its highest singlet oxygen generation and effective GA targeting features among those AIEgens, TPE-PyT-CPS significantly inhibited the tumor growth of mice without obvious adverse effects on normal tissues in vivo after PDT. It was noteworthy that on the premise of nearly equal ROS production capacity, the therapeutic effect of TPE-T-CPS is better than

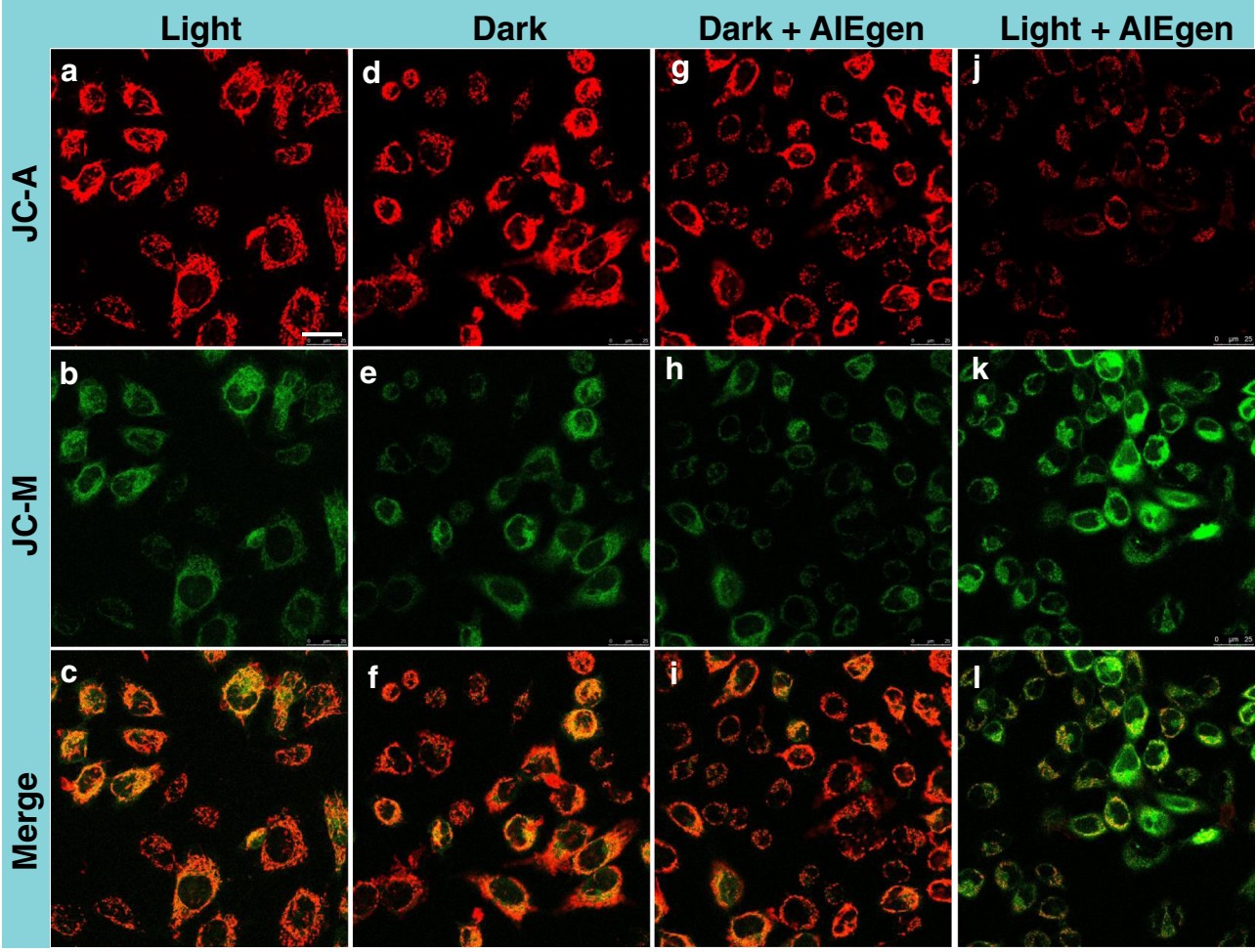

**Fig. 8 Changes of mitochondrial membrane potential after PDT.** CLSM images when HeLa cells performed with different conditions: (**a**–**c**) with light irradiation; (**d**–**f**) without light irradiation (control group); (**g**–**i**) in the presence of TPE-PyT-CPS (0.2 μM) without light irradiation; (**j**–**l**) in the presence of TPE-PyT-CPS (0.2 μM) with light irradiation (25 mW cm$^{-2}$, 2 min). JC-1 was used as a mitochondrial membrane potential indicator. The experiments were performed three times independently, representative images are shown. Scale bar: 25 μm.

TPE-PyT-PS both at the cellular and in vivo level, which was attributed to the effective GA targeting ability of TPE-T-CPS. This work provided a reliable design strategy for the development of AIEgen based GA targeting PSs, which offered a bright avenue for precise and efficient PDT through transferring stress signals from Golgi apparatus to mitochondria.

## Methods

**Materials and instrument**. All chemicals for compounds synthesis were commercially available (Sigma-Aldrich, J&K, Sinopharm, Bidepharma-Tech) and used without further purification. The silica gel (300-400 mesh) was used for column chromatography. The $^1$H and $^{13}$C NMR spectra were recorded on a Bruker DRX 400 MHz spectrometer. High-resolution mass spectra (HR-MS) were determined using an Agilent 6540Q-TOF HPLC-MS spectrometer. UV-visible absorption spectra were recorded on a PerkinElmer Lambda 35 spectrophotometer. Emission spectra were recorded on a FluoroMax-4 (Horiba). The transmission electron microscopy (TEM) images were obtained using a JEOL JEM-1011 transmission electron microscope (Japan). Fluorescence confocal imaging was carried out on a laser scanning confocal imaging system (Olympus TH4-200) consisting of ZEISS Laser Scanning Microscope (LSM 710) and a 20-mW output 488 nm argon-ion laser. Flow cytometry analysis was performed with a BD LSRFortessa Cell Analyzer. The in vivo imaging of mice was performed with PerkinElmer IVIS Lumina K Series III in vivo imaging system.

**Cell lines**. The human cervical cancer cell line HeLa were purchased from Stem Cell Bank, Chinese Academy of Science (Shanghai, China) and cultured in Dulbecco's Modified Eagle Medium (DMEM, KeyGEN) supplemented with 10% fetal bovine serum (FBS, HyClone) and 1% antibiotics, respectively, with 21% O$_2$ and 5% CO$_2$ at 37 °C. Before an experiment, the cells were passaged three times.

**Animals**. BALB/C-*nu* female mice aged 6-8 weeks of specific pathogen-free (SPF) grade were purchased from GemPharmatech Co., Ltd., license No.: SCXK(Jiangsu) 2018-0008. The mice were housed in animal-holding units in a pathogen-free environment with temperature at 22 ± 2 °C and 55 ± 5 % humidity, under the 12 h/12 h dark/light cycle. During the study, animals were observed for any clinically relevant abnormalities daily or once every other day. If any animal was moribund due to treatment-associated toxicity, tumor over-growth (≥1500 mm$^3$), loss of 20% of body weight relative to the start of the study, or the appearance of large or open ulceration in the xenograft before scheduled killing, it was killed by CO$_2$ inhalation. All the animal experiments in this study were approved by the Institutional Animal Care and Use Committees of GemPharmatech, approval No.: SYXK (Jiangsu) 2018-0027.

**Density functional theory (DFT) calculation**. Theoretical calculations were adopted to rationalize the ISC process of the TPE-PyT-CPS and TPE-T-CPS using Gaussion 09w. The geometries of TPE-PyT-CPS and TPE-T-CPS were optimized based on the method TD-DFT//B3LYP/6-31 G(d). TD-DFT was used to predict the excitation energies for the singlet and triplet excited states of TPE-PyT-CPS and TPE-T-CPS in tetrahydrofuran.

**Morphology and size**. Dynamic light scattering (DLS) was used to monitor the size of those AIEgens in water, acetonitrile solution and the morphology was observed by Transmission Electron Microscope (TEM).

**Photostability of the AIEgens**. We tested the photostability of AIEgens (5 μM) and compared them with the traditional photosensitizers including ICG

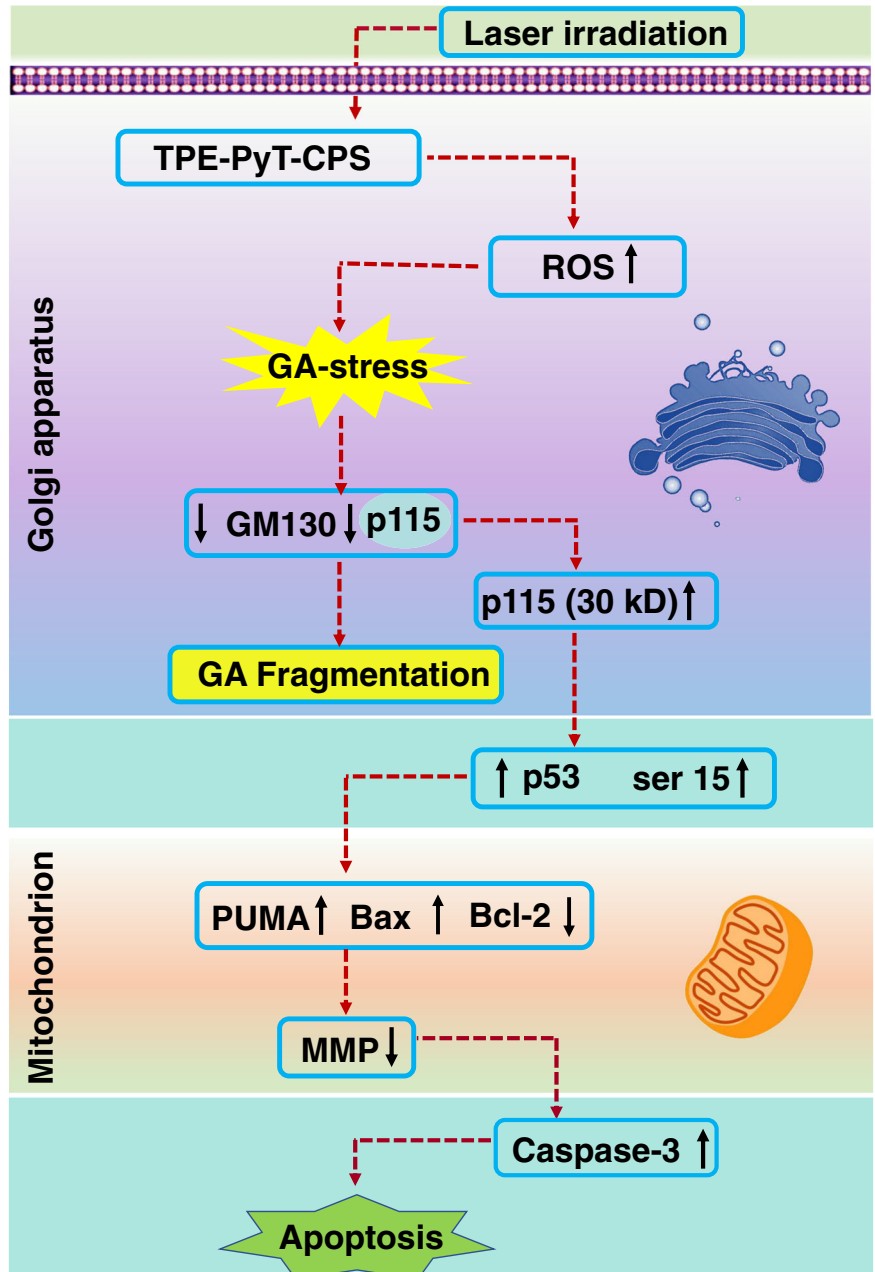

**Fig. 9 Golgi oxidative stress leads to the decrease of mitochondrial membrane potential, which leads to apoptosis.** Proposed mechanism for the cell apoptosis through the crosstalk between Golgi apparatus and mitochondria in GA-targeted photodynamic therapy. ↑: represents the increased concentration or up-regulated expression. ↓: represents the decreased value or down-regulated expression.

(Indocyanine green, 5 μM) and RB (Rose Bengal, 5 μM) in water. Due to the absorption spectra of each photosensitizer are different, we use 532 nm laser (25 mW cm$^{-2}$ for 30 min) to test the photostability of AIEgens and RB, and then 808 nm laser (25 mW cm$^{-2}$ for 30 min) to test the photostability of ICG.

**Singlet oxygen detection in solution**. ABDA (9,10-Anthracenediyl-bis(methylene)-dimalonic acid) was used as an indicator to detect singlet oxygen production capacity since the absorbance of ABDA decreases upon reaction with singlet oxygen, and RB (Rose Bengal, 10 μM) was used as an internal reference. For singlet oxygen detection, the ABDA (60 μM) was mixed with the AIEgens (10 μM) in acetonitrile/water (1:99, *v/v*) and exposed to 532 nm laser irradiation (10 mW cm$^{-2}$). The decomposition of ABDA was monitored by the absorbance decrease at 370 nm.

**ROS type evaluation**. Superoxide anion probe (HPF, 5 μM, λex = 490 nm, λem = 515 nm) and hydroxyl radical probe (DHR123, 5 μM, λex = 488 nm,

λem = 525 nm) were used to study whether TPE-PyT-CPS (1 μM) has the properties of type I photosensitizer under the irradiation of 532 laser for 300 s. Meanwhile, crystal violet (CV, 1 μM) as a previously reported type I PS was selected as the reference.

**Lipophilicity of AIEgens**. Solutions of different AIEgens (concentration:15, 30, 45 μM) were prepared in phosphate buffer (10 mM, pH 7.4) pre-saturated with 1-octanol. Equal volumes (1.5 mL) of the solution and 1-octanol pre-saturated with the phosphate buffer were mixed and placed in a thermostatic (25.0 ± 0.1 ºC) air bath orbital shaker at 200 rpm for 4 h. The samples were separated into two phases after centrifugation at 2500 rpm for 15 min. The concentration of the solute in the aqueous phase was determined by spectrophotometry. According to the law of mass conservation, the drug concentration of the 1-octanol phase and the lipo-hydro partition coefficient $P_{O/W}$ ($P_{O/W} = C_O/C_W = A_O/A_W$, A stands for absorbance) were calculated. The average of three parallel experimental data was reported as the final result. The log $P_{O/W}$ was calculated using the following

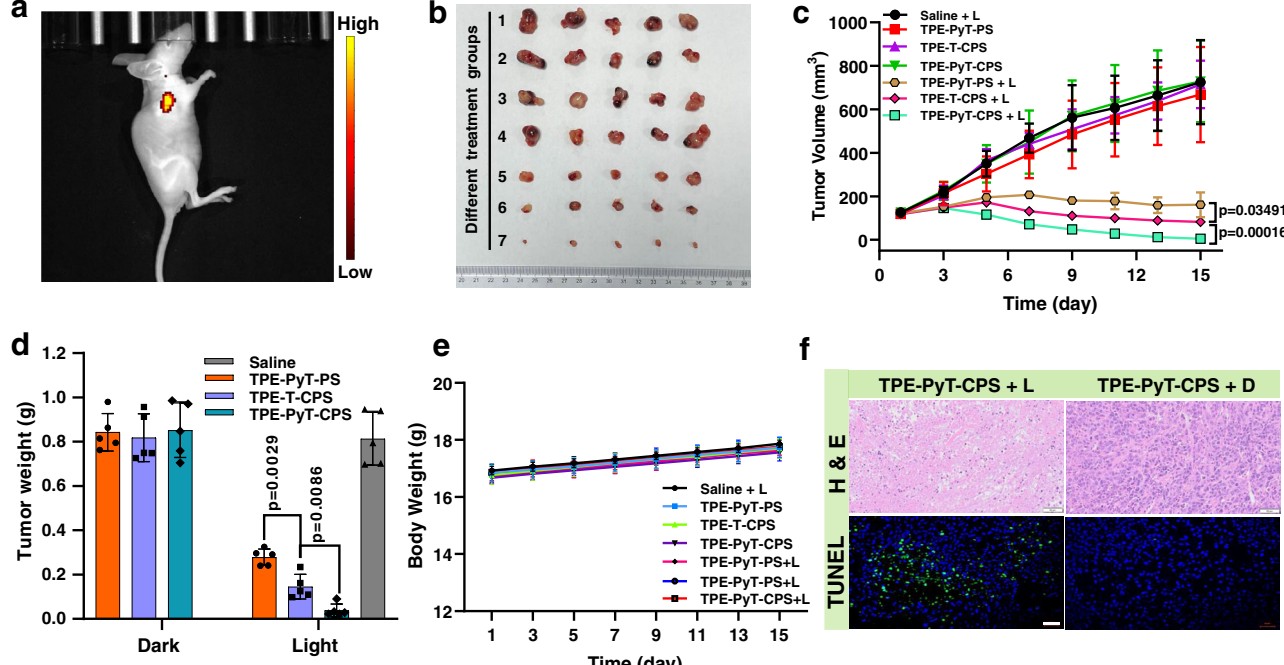

**Fig. 10 Therapeutic effect of different AIEgens in vivo.** (**a**) Fluorescent image of mice after intratumoral injection of TPE-PyT-CPS (0.1 mM, 120 μL) for 18 h. (**b**) Photographs of excised tumors on the 15th day in different groups (1. saline + L, 2.TPE-PyT-PS, 3.TPE-T-CPS, 4.TPE-PyT-CPS, 5.TPE-PyT-PS + L, 6.TPE-T-CPS + L, 7.TPE-PyT-CPS + L). (**c**) Changes of tumor volume and weight (**d**) during PDT that treated with different AIEgens (0.2 mM, 100 μL in saline) as the PS under 532 nm laser irradiation of 35 mW cm$^{-2}$ for 5 min ($n = 5$ animals). (**e**) Body weight curves of mice during PDT treatment in different treating groups ($n = 5$ animals). (**f**) H&E staining (scale bar: 50 μm) and apoptotic analysis of TUNEL staining (scale bar: 20 μm) images of tumor slices from mice after treatment with saline and TPE-PyT-CPS under 532 laser irradiations. The experiments in 10a and 10 f were performed from biologically independent animals ($n = 3$), representative images are shown. Data in 10c, d, e are presented as mean ± SD ($n = 5$ animals). Statistically significant differences between the experimental groups were analyzed by two-tailed Student's t-test. when $p < 0.05$, it was considered to have statistical significance. Source data are provided as a Source Data file.

equation (Eq. 1):

$$\log P_{O/W} = \log[([C]_{\text{initial}} - [C]_{\text{final}})/[C]_{\text{final}}] \tag{1}$$

**Cellular uptake pathways**. The cell internalization pathways of the AIEgens were investigated by different blockers including tetraethylammonium (TEA, 1 mM) was used as organic cation transporters inhibitor, 2-deoxy-D-glucose (2-DDG, 30 mM) was used as ATP synthesis inhibitor, chlorpromazine (CHP, 10 μM) was used as clathrin-mediated endocytosis inhibitor, genistein (GEN, 200 μM) was used as caveolae-mediated endocytosis inhibitor, methyl beta-cyclodextrin (MβCD, 2 mM) was used as lipid raft mediated endocytosis inhibitor, cytochalasin B (CytB, 100 μM) was used as inhibitor of macropinocytosis. First, the HeLa cells are pre-incubated in DMEM for 1 h at 37 °C with different pathway inhibitors, then the medium is replaced with different AIEgens (10 μM) including TPE-PyT-CPS, TPE-PyT-PS, TPE-PyT-CP and TPE-T-CPS. Finally, fluorescence imaging is performed using confocal microscope using 60× objective lens (Ex = 488 nm, Em = 600 − 750 nm) for those AIEgens.

**Subcellular organelle imaging**. The HeLa cancer cells were used in the following experiments and cultured using DMEM culture medium containing 10% FBS and 1% penicillin-streptomycin in an artificial environment (5% CO$_2$ at 37 °C). The HeLa cells were regularly checked for mycoplasma contamination and then conducted when the cells were grown to 80% confluence in the culture dish. The HeLa cancer cells were cultured in the special confocal chambers at a density of $10^5$ cells mL$^{-1}$ in culture medium. After 24 h, when cells have reached the desired confluence TPE-PyT-CPS (10 μM), TPE-PyT-CP (10 μM), TPE-T-CPS (10 μM) and TPE-PyT-PS (10 μM) in FBS free medium were added and incubated with HeLa cancer cells at 37 °C for 6 h.

**Mitochondrial Imaging**. The cells were then washed three times carefully using 1×PBS, followed by addition of commercial Mito Tracker Green$^{FM}$ (50 nM) and incubation for another 30 min. Subsequently, the cells were washed three times using 1×PBS and visualized by confocal laser scanning microscopy at 488 nm excitation for TPE-PyT-CPS, TPE-PyT-CP, TPE-T-CPS, TPE-PyT-PS (10 μM) and 488 nm excitation for Mito Tracker Green$^{FM}$, respectively. The signal collections of

CLSM were 600–750 nm for different AIEgens and 495–535 nm for Mito Tracker Green.

**Lysosome imaging**. The cells were then washed three times carefully using 1×PBS, followed by addition of commercial LysoSensor$^{TM}$ Green DND-189 (75 nM) and incubation for another 60 min under growth conditions appropriate for the particular cell type. Subsequently, the cells were washed three times using 1×PBS and visualized by confocal laser scanning microscopy at 488 nm excitation for TPE-PyT-CPS, TPE-PyT-CP, TPE-T-CPS, TPE-PyT-PS (10 μM) and 488 nm excitation for LysoSensor$^{TM}$ Green DND-189, respectively. The signal collections of CLSM were 600–750 nm for different AIEgens and 495-535 nm for LysoSensor$^{TM}$ Green DND-189.

**Golgi apparatus imaging**. The cells were then washed three times carefully using 1×PBS, followed by addition of commercial Golgi Green (5 μM) and incubation for 30 min under 4 °C. After, washing the cells three times using 1×PBS growth and incubation for another 30 min under 37 °C. Subsequently, visualized by confocal laser scanning microscopy at 488 nm excitation for TPE-PyT-CPS, TPE-PyT-CP, TPE-T-CPS, TPE-PyT-PS (10 μM) and 488 nm excitation for Golgi Green, respectively. The signal collections of CLSM were 600-750 nm for different AIEgens and 495-535 nm for Golgi Green.

**Endoplasmic reticulum imaging**. The cells were then washed three times carefully using 1×PBS, followed by addition of commercial ER-Tracker$^{TM}$ Blue (1 μM) and incubation for another 45 min under growth conditions (37 °C) appropriate for the particular cell type. Subsequently, the cells were washed three times using 1×PBS and visualized by confocal laser scanning microscopy at 488 nm excitation for TPE-PyT-CPS, TPE-PyT-CP, TPE-T-CPS and TPE-PyT-PS (10 μM) and 405 nm excitation for ER-Tracker$^{TM}$ Blue, respectively. The signal collections of CLSM were 600-750 nm for different AIEgens and 455–520 nm for ER-Tracker$^{TM}$ Blue.

**Quantitative analysis of photosensitizer in organelles**. We used the organelles separation kit (BestBio) to quantitatively analyze the distribution of TPE-PyT-CPS in each organelles including mitochondrial, lysosome, Golgi apparatus and endoplasmic reticulum. The experimental process is as follows: firstly, we incubated

Hela cells with TPE-PyT-CPS for 6 h and then collected cells into PE tube (2 ml), then separated each subcellular organelle according to the organelles separation kit instructions. Next, we lysed each subcellular organelle with organic solvent (methanol/chloroform = 1:1, containing 5 μL acetic acid/mL)[57] to obtain the lysis solution containing compound TPE-PyT-CPS, and finally tested the absorption in different subcellular organelle lysates with UV.

**Intracellular ROS detection**. The capability of TPE-PyT-CPS, TPE-T-CPS and TPE-PyT-PS for intracellular ROS production were assessed using singlet oxygen sensor green (SOSG) as an ROS indicator. The HeLa cancer cells were cultured in the special confocal chambers and incubated with TPE-PyT-CPS (0.2 μM), TPE-T-CPS (0.4 μM) or TPE-PyT-PS (0.4 μM) for 6 h at 37 °C. Subsequently, the cells were washed three times with 1×PBS, followed by addition of singlet oxygen sensor green (SOSG, 15 μM) in FBS free culture medium and incubation for 30 min. The above-mentioned process was performed in dark. Then, the cells were exposed to 532 nm laser irradiation (25 mW cm$^{-2}$) for 2 min, followed by imaging with CLSM. For CLSM imaging, the excitation of SOSG was 488 nm with a collection of fluorescence signal at 525 ± 20 nm.

**Cytotoxicity Study**. Briefly, 100 μL of HeLa cell suspension were added into each well of 96-well plate at a density of 8×10$^4$ cells mL$^{-1}$. After 80% confluence, fresh culture medium (the concentration of DMSO lower than 0.5%) containing a series of concentrations of AIEgens (TPE-PyT-PS, TPE-T-CPS, TPE-PyT-CPS) were added and incubated with the HeLa cancer cells for 6 h at 37 °C in dark. Subsequently, the cells were exposed to 532 nm laser irradiation (65 mW cm$^{-2}$) for 2 min. Alternatively, the different AIEgens treated HeLa cancer cells were kept in dark without light exposure. At 24 h postirradiation, the wells were replaced with freshly prepared 3-(4,5-dimethylthiazol-2-yl)-2,5-diphenyl tetrazolium bromide (MTT, 2.5 mg mL$^{-1}$ in PBS) solution. After 4 h incubation, the solution in each well was carefully removed and 150 μL of DMSO was added to each well to dissolve the formazan. The plate was gently shaken for 10 min at room temperature and then the absorbance of MTT at 595 nm was monitored by the microplate reader (Thermo Scientific Varioskan Flash) in order to determine the cell viability. Cell viability rates (%) and IC$_{50}$ values were calculated on the data of three parallel tests.

**Annexin V-FITC/Propidium Iodide Co-Staining Assay**. HeLa cells pretreated with TPE-PyT-CPS (0.2 μM) for 30 min were stained with both annexin V-FITC and PI following the standard protocol (Life Technologies) and imaged after irradiation of 532 nm laser (65 mW cm$^{-2}$, 2 min). For control experiments, HeLa cells preincubated with TPE-PyT-CPS (0.2 μM), annexin V-FITC, and PI were imaged without light irradiation. The excitation wavelengths for annexin V-FITC and PI were 488 and 559 nm, respectively, and the emissions for annexin VFITC and PI were collected in the range of 500 − 530 and 600 − 630 nm, respectively.

**Flow Cytometry**. HeLa cells were seeded in a 6 cm plate (Corning) at a density of 2×10$^5$ cells mL$^{-1}$ and incubated overnight to 70–80% confluence. Then the culture medium was replaced with medium containing TPE-PyT-CPS (0.2 μM). Photo-irradiation was performed with a 532 nm laser (65 mW cm$^{-2}$, 2 min) after 6 h of incubation. Apoptosis assay was performed 24 h after irradiation. Cells were detached by trypsin and washed with PBS. The cell precipitation was suspended into the 1× binding buffer (100 μL) containing 5 μL Annexin V-FITC (BD Biosciences) and incubated at room temperature for 1 h in the dark. The analysis was performed by using BD FACSCalibur flow cytometer within 1 h after adding 400 μL 1 × binding buffer. FlowJo_V10.6.2 was used to analyses the Flow Cytometry experiment.

**Detection the expression of p115, GM130, P53, ser15, PUMA, Bcl-2, Bax and Caspase-3**. The HeLa cancer cells were seeded into 6 cm plates at the density of 1×10$^5$ cells mL$^{-1}$. After adherence, the HeLa cancers cells were incubated with 0.2 μM of TPE-PyT-CPS, for 6 h at 37 °C, respectively. For the dark control group, there is no need to light, and for the light control group, there is no need to add photosensitizer, only light for 20 min (25 mW cm$^{-2}$). Next, the treated cells were washed and irradiated by 532 nm laser irradiation for 4, 6, 8, 10, and 20 min. After 6 h, the cells were collected and then conducted with centrifugation using 12,000 rpm for 15 min at 4 °C. Protein concentrations were measured using protein assay reagents, and equal amounts of protein per lane were separated on SDS-PAGE gel and transferred to a PVDF membrane. The membrane was incubated with Anti-p115-RhoGEF (Abcam, ab223759, 1:100 dilution), Anti-GM130 (Abcam, ab52649,1:1500 dilution), Anti-p53 (Abcam, ab26, 1:2000 dilution), Anti-p53 (phospho S15) (Abcam, ab1431, 1:1000 dilution), Anti-PUMA (Abcam, ab33906, 1:2000 dilution), Bcl-2 (Abcam, ab182858, 1:200 dilution), Bax (Abcam, ab32503, 1:5000 dilution), caspase-3 (Abcam, ab184787, 1:200 dilution) and Anti-GAPDH (Abcam, ab8245, 1:10000 dilution) followed by incubation with the peroxidase-labeled goat anti-rabbit HRP secondary antibody (Cell Signaling Technology, Anti-rabbit IgG, 7074 S, 1:2000 dilution). Western blots were visualized by enhanced chemiluminescence detection system. Relative grayscale was calculated by ImageJ (Java 1.8.0_172).

**Mitochondrial membrane potential (MMP) assay**. MMP was evaluated by confocal imaging via JC-1 staining. HeLa cells were seeded in a glass bottom cell culture dish at 40% confluence and cultured in media containing TPE-PyT-CPS (0.2 μM) respectively. After the incubation of 6 h, then cells were stained with JC-1 (Beyotime Biotechnology) following the manufacture's protocol. After rinse three times with incomplete culture medium, the cells irradiated with a laser of 532 nm (25 mW cm$^{-2}$, 2 min), and confocal imaging was immediately carried out with Leica SP8. The imaging band path for green channel of JC-M was 520-550 nm (λ$_{ex}$, 490 nm), while that for red channel of JC-A was 570-640 nm (λ$_{ex}$, 525 nm).

**In vivo fluorescent imaging**. In vivo fluorescent imaging of TPE-PyT-CPS in HeLa tumor-bearing Balb/c-nu female mice was tested by using the IVIS Lumina III in vivo Imaging System (PerkinElmer). Nude mice at 6–8 weeks old were purchased from GemPharmatech Co., Ltd., license No.: SCXK(Jiangsu) 2018-0008. TPE-PyT-CPS (100 μM, 120 μL) was injected intratumorally and fluorescent images were captured by the Lumina III at 0, 3, 6, 12, 18 and 30 h after injections (Ex: 530 nm, Em:690 nm). The mice were sacrificed at designated time points (24 h) and the tissues including heart, liver, spleen, lung, kidney, were excised and imaged by IVIS Lumina III fluorescence imaging system.

**In vivo PDT**. Tumor (HeLa) bearing Balb/c-nu female mice (tumor volume: 119.05 ± 5.80 mm$^3$) were randomly divided into seven groups (n = 5): saline group with laser irradiation group (saline + L), AIEgens groups that without laser irradiation (TPE-PyT-PS, TPE-T-CPS, TPE-PyT-CPS) and AIEgens with laser irradiation (TPE-PyT-PS + L, TPE-T-CPS + L, TPE-PyT-CPS + L). Then the photosensitizers (0.2 mM, 100 μL in saline) were injected into mice in each administration group by intratumoral injection, in which the volume of mice tumor at the time of treatment was 119.05 ± 5.80 mm$^3$. After intratumoral injection of AIEgens (0.2 mM, 100 μL) for 18 h, the tumor area of the mice was irradiated with 532 nm laser of 35 mW cm$^{-2}$ for 5 min. The tumor size was measured every 3 days and calculated as follows: volume = 1/2(tumor length) × (tumor width)$^2$.

**Histological analysis**. All mice of different groups were sacrificed on day of 15, and major organs and tumors were separated and made into slices for H&E or TUNEL staining. Major organs were collected and fixed in 4% paraformaldehyde, which were then embedded into paraffin, sliced at the thickness of 5 μm. The tissue slices were stained with H&E or TUNEL and then imaged by optical microscopy and assessed by 3 independent pathologists.

**Statistical analyses**. The statistical graphs were performed by using Microsoft Excel 2016 software (Microsoft, Redmond, WA). The results are expressed as mean ± standard deviation (SD). The statistical difference between the experimental groups were analyzed by two-tailed Student's t-test, and when the p < 0.05, it was considered to have statistical significance.

**Statistics and reproducibility**. Unless otherwise specified in figure legends, each experiment was repeated at least three times independently, and the results were similar. All images shown are representative results from biological replicates.

**Reporting Summary**. Further information on research design is available in the Nature Research Reporting Summary linked to this article.

## Data availability
The source data underlying Figs. 2a-f, 4c-e, 5b-c, 7b, 7d, 10c-e and Supplementary Figures 18a-c, 19a-f, 20a-d, 21e-h, 23a-f, 24a-j, 29a-i, 31a-b, 33, 34, 36b, 36d, 38a-b are provided as a Source Data file. All other data supporting the findings of this study are available either in the article and/or its Supplementary Information files, and can also be obtained from the authors. Source data are provided with this paper.

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

## Acknowledgements

The work was under financial supports from the National Natural Science Foundation of China (22122701, Y.C.; 21907050, Y.C.; 21731004, Z.G.; 92153303, Z.G.; 21977044, W.H.; 91953201, Z.G.), the Natural Science Foundation of Jiangsu Province (BK20202004, Z.G.; BK20190282, Y.C.) and the Excellent Research Program of Nanjing University (ZYJH004, Z.G.).

## Author contributions

M.L., Y.C., W.H., and Z.G. designed the study. M.L. synthesized probes; M.L. and C.W. performed the DFT calculations. M.L., S.Y., S.J., and T.C. performed the spectroscopic and cellular experiments; M.L., Y.G. and H.Y. performed the in vivo experiments; M.L. and Y.C. co-wrote the manuscript, Y.C., R.X., and H.F. revised the manuscript. All authors discussed the results and commented on the paper. All authors have given approval to the final version of the manuscript.

## Competing interests

The authors declare no competing interests.
