## [Peer Review File · Nature Communications]

Reviewers' Comments:

Reviewer #1:

Remarks to the Author:

In the present manuscript, Minglun Liu et al. reported a series of (D- π -A)-type NIR AIEgens with different Golgi apparatus (GA) targeting and 1O_2 generation abilities. Among them, a photosensitizer, termed as TPE-PyT-CPS, showed a high 1O_2 quantum yield and excellent GA targeting ability with a Pearson's correlation coefficient (PCC) of 0.98. The GA targeting ability was contributed by the introduction of the cyano-group, while the potent 1O_2 generation ability of TPE-PyT-CPS was due to the strong ICT process and the presence of the pyrene group. Both in vitro and in vivo studies suggested the efficient inhibition of tumor cell growth by using the PS. The authors also performed comprehensive analysis toward the apoptotic pathway upon oxidative stress induced by the GA targeting photosensitizer. It was found that the GA stress could provoke mitochondria dysfunction after the treatment of PDT. The unique GA-mitochondria crosstalk was proposed to elucidate the entire apoptosis process. Indeed, the manuscript represents the first paper to report the GA-targeting PS with AIE characteristics, which is a cutting-edge discovery in the field of phototherapy. The smart molecular design strategy will greatly promote the development of GA-targeting treatment, which has been proved to be very promising by this work. In terms of the rarity of the GA-targeting PS with superior killing effect in vitro and in vivo, we recommend the publication of this nice work after a few minor revisions:

Questions and suggestions:

(1) Some minor grammatical errors need to be revised accordingly, as shown below;

"TPE-PyT-CPS can effectively inhibited (inhibit) tumour growth"

"and repeated administration without drug resistance, which are (is one of the) problems related to the use of chemotherapeutic drugs"

"generate high dosage of 1O_2 in situ to direct (directly) cause dysfunction of subcellular organelles"

"specific GA targeting AIEgen (AIEgen) based PSs have seldomly been reported"

"resulting (resulting in) up regulation of p53 and dysfunction of mitochondria and activation of apoptotic pathway"

"The stronger electron withdrawing ability of pyridinium group than -CN group lead (leads) to stronger ICT effect and more efficient ISC of TPE-PyT-PS than TPE-PyT-CP"

"benefit (benefiting) from its high singlet oxygen generation and effective GA targeting features"

The authors are suggested to check the whole paper carefully so that the paper can be more readable.

(2) Solid-state emission spectra and quantum yields of four compounds should be provided to get a clearer picture of their photophysical properties.

(3) When talking about the measurement of 1O_2 quantum yield, the applicability of the equation (Eq. 1) in the aggregate state is still under debate. Since ABDA are negatively charged, the addition of positively charged AIEgens indeed leads to the formation of complexes through electrostatic interactions, which enables the close contact between ABDA and positively charged AIEgens and thus increases the 1O_2 quantum yield. This is one of the main reasons why the electroneutral TPE-PyT-CP showed a lower 1O_2 quantum yield, whereas we can't exclude the contribution of ICT effects. In some cases, the 1O_2 quantum yield obtained through the operation of this equation can exceed 100%. The authors should think about this question carefully, regarding the applicability of the equation. Nowadays, researchers tend to use the decomposition rate of ABDA to describe the generation efficiency of 1O_2 (Wu M, Liu X, Chen H, et al. *Angewandte Chemie*, 2021, 133(16): 9175-9180.).

(4) The comparison of 1O_2 generation capability of TPE-PyT-CPS in the solution and aggregate states is unfair since the dissolved oxygen in water and organic solvents could be different.

(5) TPE-PyT-CPS forms rod-like aggregates in the aqueous solution. Confocal microscope can be used to observe the morphology of nanorod in situ.

(6) Do TPE-PyT-CPS aggregates remain rod-like structure when dispersed in DMEM? The morphology of aggregates is believed to influence the cellular internalization of TPE-PyT-CPS.

(7) Is there any difference in the internalization rate of TPE-PyT-CPS and TPE-T-CPS while performing cell imaging?

(8) Some papers about PSs can be cited properly, such as Kang M, Zhang Z, Song N, et al. *Aggregate*, 2020, 1(1): 80-106; Ji C, Lai L, Li P, et al. *Organic dye assemblies with aggregation-induced photophysical changes and their bio-applications*. *Aggregate*, 2021: e39.

Reviewer #2:

Remarks to the Author:

Comments:

In this manuscript, authors constructed AIEgen based photosensitizers TPE-PyT-CPS, which can effectively target Golgi apparatus to induce oxidative stress and activate apoptotic pathway in HeLa cells. Meanwhile, the construct TPE-PyT-CPS enhanced the singlet oxygen quantum yield, which made it a promising system in the PDT. In addition, the manuscript was presented in a logic way, and also well written. I am interested in this research, but there are still some areas in this manuscript that need to be revised. Therefore, a major revision is recommended. Some specific issues should be addressed as follows:

1. How about the stability of AIEgen based photosensitizers in vitro?
2. Golgi targeting can generally be achieved by selecting Golgi targeting groups or compounds with a suitable log P value. According to literatures (J. Am. Chem. Soc. 2013, 135, 11663-11669; Biotechnol & Histochemistry 2013, 88, 461-476), log P = 3.5 should be required for retaining in Golgi apparatus. However, there is no Golgi targeting groups in TPE-PyT-CPS, TPE-T-CPS, TPE-PyT-CP and TPE-PyT-PS AIEgens, and the log P O/W values were determined to be 1.764, 1.582, 2.707 and 2.215, respectively. As the author described that "There seemed to be no direct link between lipophilicity and GA targeting ability of these AIEgens". So, what is the Golgi targeting mechanism of these AIEgens? Understanding the targeting mechanism will provide very important guiding significance for the design of subsequent Golgi targeted diagnostic and therapeutic reagents. The authors should study the targeting mechanism in detail, not just point out that CN group played an essential role for GA targeting in this AIEgen system.
3. The description of "It was strange that TPE-Py-CPS showed minimum localization in mitochondria, although it contained a cationic pyridinium group." (in page 3) Please check the photosensitizer is "TPE-Py-CPS" or "TPE-PyT-CPS". And the reason why the photosensitizer showed minimum localization in mitochondria should be explained.
4. In Figure 4, the cellular location of photosensitizers was conducted in 6 h. How to determine the incubation time, as the photosensitizers should achieve the organelles' targeting after the cellular uptake.
5. The endocytosis mechanism of photosensitizers is unclear in the manuscript, and is the endocytosis pathway related to the GA targeting?
6. The authors showed the CLSM images of cellular location (Figure 4). However, considering the randomness of CLSM observation, other analysis is recommended to provide some quantitative data.
7. Is TPE-PyT-CPS selective for normal cells and cancer cells? Does it only target the Golgi apparatus of cancer cells and has little effect on normal cells? Please add the co-localization experiment in which normal cells and cancer cells are incubated together.
8. The authors only used SOSG and ABDA as indicators for the measurement of ROS type, the experimental results showed that TPE-PyT-CPS has high singlet oxygen production capacity, but this does not mean that there is only singlet oxygen. Please add HPF and DHR123 indicators to test whether there is type I ROS.
9. In vitro and vivo study, why did authors choose HeLa cells as disease model. Considering the therapy and the administration pathway in the manuscript, skin cancer might be a more suitable disease model.
10. In vivo study, there need a group of photosensitizer without GA targeting to support the advantage of GA targeting photosensitizer.
11. The serum biochemistry index should be test to evaluate the safety of photosensitizer in vivo.
12. The scale bar in Figure 5D is not very clear, please upload the picture again.
13. There is no statistical analysis in any graphs. It would be essential in some of them, such as Figure 10C and 10D for changes of tumor volume and weight after PDT.
14. Test are not distinct in Figure 10, Figure S19, Figure S20, Figure S25 and Figure S26. Please reupload carefully prepared pictures.
15. In vivo PDT method is unclear. Please add the solvent of TPE-PyT-CPS and the tumor size

when treatment were implemented.

Response to reviewers

Reviewer #1

Q: In the present manuscript, Minglun Liu et al. reported a series of (D- π -A)-type NIR AIEgens with different Golgi apparatus (GA) targeting and 1O_2 generation abilities. Among them, a photosensitizer, termed as TPE-PyT-CPS, showed a high 1O_2 quantum yield and excellent GA targeting ability with a Pearson's correlation coefficient (PCC) of 0.98. The GA targeting ability was contributed by the introduction of the cyano-group, while the potent 1O_2 generation ability of TPE-PyT-CPS was due to the strong ICT process and the presence of the pyrene group. Both in vitro and in vivo studies suggested the efficient inhibition of tumor cell growth by using the PS. The authors also performed comprehensive analysis toward the apoptotic pathway upon oxidative stress induced by the GA targeting photosensitizer. It was found that the GA stress could provoke mitochondria dysfunction after the treatment of PDT. The unique GA-mitochondria crosstalk was proposed to elucidate the entire apoptosis process. Indeed, the manuscript represents the first paper to report the GA-targeting PS with AIE characteristics, which is a cutting-edge discovery in the field of phototherapy. The smart molecular design strategy will greatly promote the development of GA-targeting treatment, which has been proved to be very promising by this work. In terms of the rarity of the GA-targeting PS with superior killing effect in vitro and in vivo, we recommend the publication of this nice work after a few minor revisions:

A: We appreciate the reviewer's comments, we have prepared a point-by-point response to each question as shown blow.

Q: (1) Some minor grammatical errors need to be revised accordingly, as shown below;

“TPE-PyT-CPS can effectively inhibited (inhibit) tumour growth”

“and repeated administration without drug resistance, which are (is one of the) problems related to the use of chemotherapeutic drugs”

“generate high dosage of 1O_2 in situ to direct (directly) cause dysfunction of subcellular organelles”

“specific GA targeting AIEgen (AIEgen) based PSs have seldomly been reported”

“resulting (resulting in) up regulation of p53 and dysfunction of mitochondria and activation of apoptotic pathway”

“The stronger electron withdrawing ability of pyridinium group than -CN group lead (leads) to stronger ICT effect and more efficient ISC of TPE-PyT-PS than TPE-PyT-CP”

“benefit (benefiting) from its high singlet oxygen generation and effective GA targeting features”

The authors are suggested to check the whole paper carefully so that the paper can be more readable.

A: We sincerely thank the reviewer for pointing this out. The above grammatical problems occurred in our writing have been revised according to the suggestions of

the reviewer and the revisions in the manuscript were marked in yellow color.

Q: (2) Solid-state emission spectra and quantum yields of four compounds should be provided to get a clearer picture of their photophysical properties.

A: We agree with the reviewer and now include the solid-state fluorescence spectrum and solid-state fluorescence quantum yield data to improve the photophysical properties of each compound. As shown in the **Figure R1**, the maximum solid-state emission of TPE-PyT-CP and TPE-PyT-PS is 703 nm and 704 nm, respectively. The maximum emission of TPE-T-CPS is 748 nm and that of TPE-PyT-CPS is 756 nm, which shows a slight red shift compared with TPE-T-CPS. The solid-state fluorescence quantum yields of TPE-PyT-CP, TPE-PyT-PS, TPE-T-CPS and TPE-PyT-CPS were 2.79%, 4.95%, 5.25%, 5.63% (**Table 1**), respectively.

Figure R1. Fluorescence emission spectra of each AIEgens in solid state.

Table R1. Optical properties and lipophilicity of AIEgens including TPE-PyT-CP, TPE-PyT-PS, TPE-T-CPS, TPE-PyT-CPS.

AIEgens	$\lambda_{\text{abs}}^{\text{a}}$ (nm)	λ_{em} (nm)		$\Phi_{\text{f}}^{\text{c}}$ (%)			Log $P_{\text{O/W}}$
		Water ^b	Solid state	Water	ACN	Solid state	
TPE-PyT-CP	428	625	703	12.1	1.28	2.79	2.707 ± 0.007
TPE-PyT-PS	438	653	704	13.6	0.78	4.95	2.215 ± 0.009
TPE-T-CPS	472	674	748	19.8	1.71	5.25	1.582 ± 0.006
TPE-PyT-CPS	487	680	756	17.1	2.74	5.63	1.764 ± 0.004

a) Absorption maximum in water (containing 1% acetonitrile). b) Emission maximum in water (containing 1% acetonitrile). c) Fluorescence quantum yield of AIEgens in water, acetonitrile and solid state.

Q: (3) When talking about the measurement of $^1\text{O}_2$ quantum yield, the applicability of the equation (Eq. 1) in the aggregate state is still under debate. Since ABDA are negatively charged, the addition of positively charged AIEgens indeed leads to the formation of complexes through electrostatic interactions, which enables the close

contact between ABDA and positively charged AIEgens and thus increases the $^1\text{O}_2$ quantum yield. This is one of the main reasons why the electroneutral TPE-PyT-CP showed a lower $^1\text{O}_2$ quantum yield, whereas we can't exclude the contribution of ICT effects. In some cases, the $^1\text{O}_2$ quantum yield obtained through the operation of this equation can exceed 100%. The authors should think about this question carefully, regarding the applicability of the equation. Nowadays, researchers tend to use the decomposition rate of ABDA to describe the generation efficiency of $^1\text{O}_2$ (Wu M, Liu X, Chen H, et al. *Angewandte Chemie*, 2021, 133(16): 9175-9180).

A: We sincerely thank the reviewer for this constructive comment. The negative charge of ABDA may indeed interact strongly with the positively charged photosensitizer, thus it may lead to inaccurate singlet oxygen yield. Therefore, in revised manuscript the decomposition rate of ABDA was used as the basis to evaluate the singlet oxygen generation capacity of photosensitizer. After irradiation, the decomposition rates of ABDA were 9.63, 26.97, 27.91, 32.85, 21.69 nM/min in the presence of TPE-PyT-CP, TPE-PyT-PS, TPE-T-CPS, TPE-PyT-CPS and RB (**Figure 2**), respectively.

Figure R2. Decomposition rate of ABDA in the presence of different AIEgens (10 μM). RB (A), TPE-PyT-CP (B), TPE-PyT-PS (C), TPE-T-CPS (D), TPE-PyT-CPS (E) and (F) Plot of A/A_0 against light exposure time for different AIEgens in water (containing 1% acetonitrile, v/v), where A_0 and A are the ABDA (60 μM) absorbance (370 nm) before and after irradiation, respectively.

Q: (4) The comparison of $^1\text{O}_2$ generation capability of TPE-PyT-CPS in the solution and aggregate states is unfair since the dissolved oxygen in water and organic solvents could be different.

A: We thank the reviewer for this important comment. It is true that the solubility of oxygen varies in different solvents, so we searched the literature for the solubility of oxygen in aqueous solution and acetonitrile, respectively. Oxygen solubility in pure or fresh water at 25 $^\circ\text{C}$ and 1.0 atm of O_2 pressure is about 1.22 mM and the values of

which are varied from 1.18 to 1.25 mM as reported in different literature (Xing, W.; Yin, M.; Lv, Q.; Hu, Y.; Liu, C.; Zhang, J., *Rotating Electrode Methods and Oxygen Reduction Electrocatalysts*. 2014; pp 1-31.)

We found that the solubility of oxygen in acetonitrile was 8.1 mM/L (Li, Q.; Batchelor-McAuley, C.; Lawrence, N. S.; Hartshorne, R. S.; Compton, R. G., *J Electroanal Chem*. 2013, 688, 328-335; Achord, J. M.; Hussey, C. L., *Anal. Chem*. 1980, 52, 601–602). This is reasonable that the solubility of apolar O₂ in water was poorer than that in acetonitrile since the polarity of water is higher than acetonitrile. Therefore, our data and conclusion should be correct, the singlet oxygen production capacity of the AIEgens in the aggregated state is higher than that in the dissolved state.

Q: (5) TPE-PyT-CPS forms rod-like aggregates in the aqueous solution. Confocal microscope can be used to observe the morphology of nano-rod *in situ*.

A: We thank the reviewer for this important comment. As shown in **Figure R3**, we observed rod-like aggregates in aqueous solution though the confocal imaging.

Figure R3. Aggregates of compound TPE-PyT-CPS (20 μM) in aqueous solution observed by confocal microscope.

Q: (6) Do TPE-PyT-CPS aggregates remain rod-like structure when dispersed in DMEM? The morphology of aggregates is believed to influence the cellular internalization of TPE-PyT-CPS.

A: Thanks to the reviewer for this important comment. The aggregation of compound TPE-PyT-CPS was observed in DMEM medium via confocal imaging. As shown in the **Figure R4**, the TPE-PyT-CPS was distributed in the DMEM medium in rod-like shape with different size. In addition, we found that those AIEgens can form aggregates of different sizes in aqueous solution (**Figure R11**), so we studied their cell entry rate through cell uptake experiment (**Figure R5** and **R6**). Just as the reviewer said, the morphology of the aggregates indeed distinctly influences the cellular uptake of AIEgens.

Figure R4. Aggregates of compound TPE-PyT-CPS (20 μM) in aqueous solution observed by confocal microscope.

Q: (7) Is there any difference in the internalization rate of TPE-PyT-CPS and TPE-T-CPS while performing cell imaging?

A: We thank the reviewer for the important questions. The amount of photosensitizer uptake by cells is very important for phototoxicity, so we studied the changes of AIEgens uptake by HeLa cells over time. As shown in the **Figure R5**, the uptake of TPE-PyT-CPS, TPE-PyT-CP and TPE-T-CPS by cells increased gradually in 0-3 hours, but the intensity was weakly in general. After co-incubation with the cells for about 6 hours, each compound was obviously distributed in the cells, and the uptake changed little with the passage of time (**Figure R6**). The above results showed that the AIEgens of TPE-PyT-CPS, TPE-PyT-CP and TPE-T-CPS could enter HeLa cells and the uptake was close to saturation after 6 hours. Hence, there is little difference in the uptake between TPE-PyT-CPS and TPE-T-CPS at 37 $^{\circ}\text{C}$. Specifically, we found that TPE-PyT-PS was obviously distributed in HeLa cells in about 3 hours compared with other AIEgens. This may be due to TPE-PyT-PS has smaller size in solution compared with AIEgens, so it can enter the cells through the passive transport (**Figure R9, 10 and 11**).

Figure R5. The uptake of AIEgens (10 μM) by HeLa cells at different times (0-9 h). (Scale bar: 10 μm).

Figure R6. Mean fluorescence intensity (MFI) of each photosensitizer in HeLa cells at different times (0-9 h).

(8) Some papers about PSs can be cited properly, such as Kang M, Zhang Z, Song N, et al. *Aggregate*, 2020, 1(1): 80-106; Ji C, Lai L, Li P, et al. *Organic dye assemblies with aggregation-induced photophysical changes and their bio-applications.*

Aggregate, 2021: e39.

A: Thanks to the reviewer for this suggestion and we have added these new literatures in the revised manuscript and marked it with yellow background in the reference.

Reviewer #2:

Q: In this manuscript, authors constructed AIEgen based photosensitizers TPE-PyT-CPS, which can effectively targeted Golgi apparatus to induce oxidative stress and activate apoptotic pathway in HeLa cells. Meanwhile, the construct TPE-PyT-CPS enhanced the singlet oxygen quantum yield, which made it a promising system in the PDT. In addition, the manuscript was presented in a logic way, and also well written. I am interested in this research, but there are still some areas in this manuscript that need to be revised. Therefore, a major revision is recommended.

A: Thanks very much for your comments, we have provided a point-by-point response to each question.

(1) How about the stability of AIEgen based photosensitizers in vitro?

A: We thank the reviewer very much for this important question. Indeed, the stability of photosensitizers is essential for photodynamic therapy. Therefore, we tested the photostability of the AIEgens and compared them with the traditional photosensitizers ICG (Indocyanine green) and RB (Rose Bengal). Due to the absorption spectra of each photosensitizer are different, we use 532 nm laser (25 mW cm^{-2} for 30 min) to test the photostability of AIEgens and RB, and then 808 nm laser (25 mW cm^{-2} for 30 min) to test the photostability of ICG (**Figure R7**). The results show that the attenuation rate of AIEgens is less than 5% after 30 min irradiation (**Figure R8**). However, the attenuation rate of RB is 8.37 % and the ICG is more than 10 %, indicating that the photostability of AIEgens is higher than that of traditional photosensitizer (RB, ICG).

Figure R7. Photosability of different AIEgens (5 μM), ICG (5 μM) and RB (5 μM) in water under laser (AIEgens and RB: 532 nm laser, 25 mW cm^{-2} , ICG: 808 nm laser, 25 mW cm^{-2}) irradiation for 30 minutes.

Absorbance	TPE-PyT-CPS	TPE-PyT-PS	TPE-T-CPS	TPE-PyT-CP	Rose Bengal	Indocyanine Green
Maximum absorption	489 nm	429 nm	473 nm	427 nm	540 nm	770 nm
A_0	0.21277	0.13854	0.21559	0.16265	0.86336	1.59787
A	0.20785	0.13525	0.20851	0.15589	0.79108	1.38404
Attenuation rate: $A_0 - A / A$	2.31 %	2.37 %	3.28 %	4.15 %	8.37 %	13.38 %

Figure R8. Attenuation rate of AIEgens, RB and ICG after laser irradiation for 30 min (A_0 : Photosensitizer before irradiation; A: Photosensitizer after irradiation).

Q: (2) Golgi targeting can generally be achieved by selecting Golgi targeting groups or compounds with a suitable log P value. According to literatures (*J. Am. Chem. Soc.* 2013, 135, 11663-11669; *Biotechnic & Histochemistry* 2013, 88, 461-476), log P = 3-5 should be required for retaining in Golgi apparatus. However, there is no Golgi targeting groups in TPE-PyT-CPS, TPE-T-CPS, TPE-PyT-CP and TPE-PyT-PS AIEgens, and the log P_{O/W} values were determined to be 1.764, 1.582, 2.707 and 2.215, respectively. As the author described that "There seemed to be no direct link between lipophilicity and GA targeting ability of these AIEgens". So, what is the Golgi targeting mechanism of these AIEgens? Understanding the targeting mechanism will provide very important guiding significance for the design of subsequent Golgi targeted diagnostic and therapeutic reagents. The authors should study the targeting mechanism in detail, not just point out that CN group played an essential role for GA targeting in this AIEgen system.

A: We thank the reviewer for the comments and this is actually a VERY good question. Indeed, two kinds of Golgi-targeting groups have been reported, including Golgi-targeting polypeptides and cysteine [*J. Am. Chem. Soc.*, 2010, 132, 4455–4465, *Chem. Sci.* 8 (2017) 6829–6835, *Chem. Sci.* 10 (2019) 879–883]. Log P value is an important parameter for predicting the membrane penetration ability of a compound [*Chem. Rev.* 1993, 93, 1281–1306]. There is an empirical model of whether dyes can effectively target subcellular organelles, namely QSAR (quantitative structure-activity relationship) models. According to the model, the range of log P value for targeting different subcellular organelles are as following: endoplasmic reticulum (ER): 0-6, mitochondrion: 0-5, lysosome: >8; Golgi apparatus: 0-8 (**Table R2**) [*Biotechnic & Histochemistry* 2015, 90(4): 241–254]. Moreover, we did not find that the log P range of 3-5 is responsible for targeting Golgi in the literature of "Biotechnic & Histochemistry 2013, 88(8): 461–476" that cited by article of "*J. Am. Chem. Soc.* 2013, 135, 11663-11669".

For the AIEgens we synthesized in this paper, their log P value range is from 1.5 to 2.8,

which obeys the rule of targeting GA in the QSAR model, but TPE-PyT-PS did not target GA. From this regard, we think the dyes target to Golgi is not only determined by the value of log P.

Table R2 Decision-logic table illustrating influences of probe physicochemistry on intracellular localizations of fluorescent probes in live cells.

If	And if	And if	And if	And if	Or if	Then probe is ...	Source of data or model
$5 > \log P > 0$	$AI < 3.5$	$Z = 0$				Localized in cytosol	D'Souza et al. (2008)
$5 > \log P > 0$	$Z < 0$	$pK_a < 2$				Localized in cytosol	Rashid (1991)
$6 > \log P > 0$	$6 > AI > 3.5$	$Z > 0$				Localized in endoplasmic reticulum	Colston et al. (2003)
$\log P > 5$	$AI < 3.5$	$pK_a < 6$ [if $Z > 0$]				Localized in lipid droplets	Christensen et al. (1999), Horobin (2010)
$8 > \log P > 5$						Localized in generic biomembranes	Christensen et al. (1999), Horobin (2010)
$8 > AI > 5$						Localized in generic biomembranes	Horobin (2001)
$8 > \log P > 0$	$8 > AI > 3.5$	$Z = 0$				Localized in Golgi membranes	Rashid-Doubell & Horobin (1993), Horobin et al. unpublished results
$0 > \log P > -5$	$10 > pK_a > 6$	$Z > 0$				Localized in lysosomes/acidic organelles by ion-trapping	Rashid et al. (1991)
$\log P_{\text{free acid}} > 0$	$pK_a = 7 \pm 3$	$Z < 0$ Most-ionized species				Localized in lysosomes/acidic organelles by precipitation trapping	Rashid et al. (1991), Gilroy et al. (1991)
$\log P < 0$	$pK_a >> 7$	$CBN < 40$			$pK << 7$	Localized in lysosomes/endosomes by pinocytotic uptake	Rashid et al. (1991)
$\log P < -10$	$pK_a >> 7$	$CBN > 40$			$pK << 7$	Localized in lysosomes/endosomes by pinocytotic uptake	Rashid et al. (1991)
$\log P$ Most ionized species < -5	$pK_a = 7$	$CBN < 40$				Localized in lysosomes/endosomes by pinocytotic uptake	Rashid et al. (1991)
$AI > 8$						Localized in lysosomes/endosomes by endocytosis, lipid-bound	Christensen et al. (1999)
$\log P > 8$						Localized in lysosomes/endosomes by endocytosis, lipid-bound	Rashid et al. (1991), Christensen et al. (1999)
$CBN > 40$						Localized in lysosomes/endosomes by endocytosis, protein-bound	Rashid et al. (1991)

To elucidate the GA-targeting mechanism, we used different biochemical inhibitors including ATP synthesis inhibitor (2-deoxy-D-glucose, 2-DDG), clathrin-mediated endocytosis inhibitor (chlorpromazine, CHP), caveolae-mediated endocytosis inhibitor (genistein, GEN), lipid raft mediated endocytosis inhibitor (M β CD), macropinocytosis inhibitor (cytochalasin B, CytB) and organic cation transporters inhibitor (TEA) to study the uptake of AIEgens in HeLa cells. As shown in **Figure R9**, the results demonstrate that there is no distinct fluctuation in cell internalization percentages for TPE-PyT-PS after treated with all of the blockers. Moreover, the internalization rate of TPE-PyT-PS was close to saturation after incubated with HeLa cells for 3 hours, while other AIEgens reaches saturated after 6 hours (**Figure R5, R6**). The energy-independent manner of entering cells combined with the smaller particle size (7.2 nm) (**Figure R11**) that endows TPE-PyT-PS a faster internalization rate makes us assumed that the pattern of TPE-PyT-CPS enters cells is passive diffusion (**Figure R9, R10**). Meanwhile, the uptake of TPE-PyT-CPS with GA-targeting ability significantly decreased by 82.9% under the inhibition of 2-DDG, suggesting its energy-dependent uptake manner. In addition, the uptake of TPE-PyT-CPS manifest a decrease of 65.8% and 55.5% in HeLa cells, respectively, in the presence of GEN and M β CD, indicating their enter cells via a caveolin/raft mediated endocytosis and the same endocytosis manner exist in TPE-T-CPS and TPE-PyT-CP. Caveolin is an

essential protein component required for the formation of caveolae on the plasma membrane (*Molecular Biology of the Cell*, 2006, 17, 3085-3094) and caveolae are abundant cell-surface organelles involved in lipid regulation and endocytosis (*Cell*, 2008, 132, 113-124). Specially, studies have shown that small molecules can target the golgi apparatus via the caveolin/raft mediated endocytic pathways [Le, P. U.; Nabi, I. R., *J Cell Sci* 2003, 116 (6), 1059-1071; Pang, H.; Le, P. U.; Nabi, I. R., *J Cell Sci.* 2004, 117 (8), 1421-1430, Tan, W.; Zhang, Q.; Wang, J.; Yi, M.; He, H.; Xu, B., *Angew Chem Int Ed.* 2021, 60 (23), 12796-12801]. Hence, caveolin /raft mediated endocytosis patterns should be the reason why TPE-PyT-CPS can effectively target Golgi apparatus.

Figure R9. Fluorescence images of HeLa cells incubated with the different AIEgens (10 μ M) in the presence of different cell uptake inhibitors including 2-DDG, CHP, GEN, M β CD, CytB and TEA.

Figure R10. Mean fluorescence intensity of HeLa cells co-cultured with AIEgens for 6 hours and then treated with different pathway inhibitors including 2-DDG, CHP, GEN, MβCD, CytB and TEA (n = 4 biologically independent cells, ***p < 0.001).

To better understand the different uptake manners between TPE-PyT-PS and other AIEgens, we then studied their morphology in aqueous solution using transmission electron microscope (TEM). As shown in **Figure R11**, TPE-PyT-PS forms small spherical particles in aqueous solution with the size about 7.2 nm. In addition, TPE-PyT-PS has higher lipophilic (Log P, 2.215) that makes it easier to interact with lipid cell membranes, and combines its smaller size, making it entering cells by passive diffusion. On the other hand, other AIEgens exhibited rod-shaped structures with a size from 200 nm to 400 nm, which may be due to the highly polar cyano group can improve the order of molecular arrangement by affecting the molecular orientation (Kubicki, M., *Acta Crystallogr C*, 2004, 60 (4), 255-257; Shimizu, M.; Nata, M.; Watanabe, K.; Hiyama, T.; Ujiiie, S. *Mol. Cryst. Liq. Cryst*, 2005, 441 (1), 237-241; Alaasar, M.; Tschierske, C., *Liquid Crystals*, 2019, 46 (1), 124-130). Therefore, TPE-PyT-CPS, TPE-T-CPS and TPE-PyT-CP entering the cells with manner of caveolin/raft mediated endocytosis making them better GA targeting dyes. –CN group indeed played crucial role since it is responsible for stronger intermolecular interactions and forming larger aggregates, which ultimately lead to GA targeting. Thanks again for pushing us this far, leading to better understanding of the GA targeting mechanism of the AIEgen based PSs.

Figure R11. (A) Dynamic light scattering (DLS) and (B) transmission electron microscopy (TEM) images of different AIEgens.

Q: (3) The description of “It was strange that TPE-Py-CPS showed minimum localization in mitochondria, although it contained a cationic pyridinium group.” (in page 3) Please check the photosensitizer is “TPE-Py-CPS” or “TPE-PyT-CPS”. And the reason why the photosensitizer showed minimum localization in mitochondria should be explained.

A: Firstly, we thank the reviewers for correcting the name of the compound in the text. It should be “TPE-PyT-CPS” instead of “TPE-Py-CPS”. Secondly, thank the reviewer for the constructive comments. Many different lipophilic cations have a sufficiently large hydrophobic surface area to permeate membranes and accumulate within mitochondria. For ATP synthesis, an electrochemical gradient (i.e., a proton gradient) between the intermembrane space (IMS) and matrix in mitochondria is required so that ATP synthase can function [L.D. Zorova et al. / *Analytical Biochemistry*, 552 (2018) 50-59]. This process makes the inner mitochondrial membrane (IMM) highly negatively polarized, leading to higher proton concentrations (i.e., higher acidity) in the IMS than in the mitochondrial matrix. The positive charge lipophilic cations can effectively accumulate in mitochondria through electrostatic interaction between the negative charge in the inner membrane and lipophilic cations. For example, fluorescent lipophilic cations, such as rhodamine, JC-1, and the MitoTracker compounds, are widely used to visualize mitochondria selectively within cells (*Annu. Rev. Cell Biol.* 1998, 4:155–81; *Biochem. Biophys. Res. Commun.* 1993, 197:40–45; *Cytometry*, 1997, 27:358–64). However, carrying a positive charge may be a necessary but insufficient condition for targeting mitochondria. For example, Tang’group designed a series of photosensitizers targeting different subcellular organelles with positively charged pyridinium moiety in 2020 (**Figure R12**) (*Angew. Chem. Int. Ed.* 2020, 59, 9610 – 9616).

Figure R12. Chemical structures of the three AIEgens with different subcellular organ targeting ability.

In the paper, TFPy can target to mitochondria because of its high efficiency of electrophoretic transmembrane migration, as well as appropriate binding ability between the positively charged pyridinium moiety and the negatively charged interior of the transmembrane potential of mitochondria (L. B. Chen, *Annu. Rev. Cell Dev. Biol.* 1988, 4, 155–181). Compared with TFPy, TPE-TFPy contains an extra triphenylethylene segment at the lipophilic part, leading to a more hydrophobic nature. TPE-TFPy tends to form nano-sized aggregates in culture media due to its high hydrophobicity, and the in situ generated aggregates can internalize into lysosome. In the case of TFVP, linking a quaternary ammonium salt tail to the pyridine moiety produces an elongated hydrophilic fragment, which greatly reduces the permeation ability of TFVP through cell membrane and thus accumulating within it. Therefore, we hypothesized that the introduction of pyrene, a large π conjugated system, into TPE-PyT-CPS would increase the hydrophobicity of the molecule, and therefore it could not effectively target mitochondria.

Q: (5) In Figure 4, the cellular location of photosensitizers was conducted in 6 h. How to determine the incubation time, as the photosensitizers should achieve the organelles' targeting after the cellular uptake.

A: We thank the reviewer for the constructive suggestions. The amount of photosensitizer uptake by cells is very important for phototoxicity, so we studied the changes of photosensitizer uptake by HeLa over time. As shown in the **Figure R5** and **R6**, the uptake of TPE-PyT-CPS, TPE-PyT-CP and TPE-T-CPS by cells increased gradually in 0-3 hours, but the intensity was weakly in general. After co-incubation with the cells for about 6 hours, each compound was obviously distributed in the cells, and the uptake changed little with the passage of time, while the time for TPE-PyT-PS to reach uptake saturation is only about 3 hours (**Figure R6**). Therefore, we chose to do co-localization experiment on each photosensitizer after 6 hours.

Q: (6) The endocytosis mechanism of photosensitizers is unclear in the manuscript, and is the endocytosis pathway related to the GA targeting?

A: We sincerely thank the reviewer for raising this question. And the answer of this question is detailed above.

Q: (7) The authors showed the CLSM images of cellular location (Figure 4). However, considering the randomness of CLSM observation, other analysis is recommended to provide some quantitative data.

A: We thank the reviewer for the constructive comment. Considering the state of cells and possible improper operation during the co-localization experiment, the results made by using confocal may deviate from the actual distribution of TPE-PyT-CPS in organelles. Therefore, we used the organelle-separation kit to quantitatively analyze the distribution of TPE-PyT-CPS in different organelles (Mitochondria, lysosome, Golgi apparatus and endoplasmic reticulum). The experimental process is as follows

(Figure R13): firstly, we incubated HeLa cells with TPE-PyT-CPS for 6 hours in 37°C, then separated each subcellular organelle according to the instructions of organelle separation kit, and next we lysed each subcellular organelle with organic solvent to obtain the lysis solution containing compound TPE-PyT-CPS, and finally tested the absorption in different subcellular organelle lysates with UV.

The results showed that deep red color appeared in the Golgi apparatus after separation by the kit compared with other organelles (Figure R14A). According to the UV absorption spectrum, the absorption curve of TPE-PyT-CPS in the lysate is consistent with that of pure TPE-PyT-CPS, and the absorption value in Golgi apparatus is more than 8 times higher than that in lysosome (Figure R14B, C). The above results confirmed that TPE-PyT-CPS can effectively target Golgi apparatus.

Figure R13. Quantitative analysis of TPE-PyT-CPS in different suborganelles using UV (Mixed solvent: methanol/chloroform = 1:1, containing 5 μ l acetic acid/ml).

Figure R14. Quantitative analysis of TPE-PyT-CPS in different suborganelles. (A) Pictures of TPE-PyT-CPS in each suborganelle after separation by different suborganelle kits. (B) and (C) UV absorption spectra of each suborganelle after solvent lysis.

Q: (8) Is TPE-PyT-CPS selective for normal cells and cancer cells? Does it only target the Golgi apparatus of cancer cells and has little effect on normal cells? Please add the co-localization experiment in which normal cells and cancer cells are incubated together.

A: The question raised by the reviewer is very important. In the process of PDT, it is highly desired to develop a safe and efficient PDT platform for PDT with enhance selectivity and reduced side effect (*RSC Adv.*, 2018, 8, 42374–42379).

Figure R15. CLSM of L02 cells stained with Golgi-Green and TPE-PyT-CPS (10 μM) (scale bar: 10 μm).

In this paper, TPE-PyT-CPS has high phototoxicity (170 nM) and negligible dark toxicity ($> 256 \mu\text{M}$) to HeLa cells by effectively targeting GA. However, we did not investigate whether TPE-PyT-CPS targets the GA of normal cells. Therefore, we conducted co-localization experiments to investigate whether TPE-PyT-CPS has targeting effect on the Golgi apparatus of L02 cells (Normal liver cells). As shown in **Figure R15**, the co-localization coefficient of TPE-PyT-CPS and Golgi-green in L02 cells was as high as 0.91. Next, we co-cultured L02 cells with HeLa cells, and then did co-localization experiments to study whether there were differences in Golgi targeting ability and cell uptake between L02 and HeLa cells. The results showed that after co-incubation L02 and HeLa cells together, the co-location coefficient of TPE-PyT-CPS and Golgi-green was 0.86, and the fluorescence intensity of TPE-PyT-CPS in the two kinds of cells was basically the same.

(9) The authors only used SOSG and ABDA as indicators for the measurement of ROS type, the experimental results showed that TPE-PyT-CPS has high singlet oxygen production capacity, but this does not mean that there is only singlet oxygen. Please add HPF and DHR123 indicators to test whether there is type I ROS.

A: We thank the reviewer for the comments and constructive suggestions. According to the mechanism of action, oxygen plays an important role in PDT. Compared with type I photosensitizer, type II photosensitizer is more dependent on oxygen in the process of PDT. Therefore, it is very helpful for TPE-PyT-CPS to have both type I and type II properties to overcome tumor hypoxia. Superoxide anion and hydroxyl radical are the active sources of type I photosensitizer, so we used superoxide anion probe (HPF) and hydroxyl radical probe (DHR123) respectively to study whether TPE-PyT-CPS has the properties of type I photosensitizer. Meanwhile, crystal violet (CV) as a previously reported type I PS was selected as the reference (K. Reszka, F. S. Cruz, R. Docampo, *Chem.Biol. Interact.* 1986, 58, 161-172). As shown in **Figure R16** and **R17**, the PL intensity in “DHR123 + Light”, “Crystal violet & DHR123 + Dark” or “HPF + Light”, “Crystal violet & HPF + Dark” group was only slight fluorescence enhancement. Similarly, the fluorescence intensity of DHR123 or HPF has a little enhancement with the increasing of irradiation time in the presence of TPE-PyT-CPS (532 nm laser, 10 mW cm^{-2}), suggesting that TPE-PyT-CPS was not capable of

generating $\cdot\text{OH}$ or $\cdot\text{O}_2^-$ effectively through type-I process. By contrast, obviously fluorescence enhancement of DHR123 or HPF was observed in the presence of CV under the same conditions (**Figure R17**), further indicating the TPE-PyT-CPS was not the type I photosensitizer.

Figure R16. (A) PL spectra of DHR123 (for $\cdot\text{O}_2^-$ detection) and (B) TPE-PyT-CPS ($1\mu\text{M}$) in the presence of DHR123 upon 532 nm laser irradiation (10 mW cm^{-2}) for different times. (C) PL spectra of HPF (for $\cdot\text{OH}$ detection) and (D) TPE-PyT-CPS ($1\mu\text{M}$) in the presence of HPF upon 532 nm laser irradiation (10 mW cm^{-2}) for different times.

Figure R17. PL spectra of crystal violet ($1\mu\text{M}$) and DHR123 without (A) or with (B) 532 nm laser

irradiation (10 mW cm^{-2}) for different times. PL spectra of crystal violet ($1 \mu\text{M}$) and HPF without (D) or with (E) 532 nm laser irradiation (10 mW cm^{-2}) for different times. Relative changes in PL intensity of DHR123 (for $\cdot\text{O}_2^-$ detection) (C) and HPF (for $\cdot\text{OH}$ detection) (F) in the presence of different photosensitizer with or without 532 nm laser irradiation (10 mW cm^{-2}).

Q: (10) In vitro and vivo study, why did authors choose HeLa cells as disease model. Considering the therapy and the administration pathway in the manuscript, skin cancer might be a more suitable disease model.

A: We thank the reviewer very much for the important comments. HeLa line, derived from the cervical cancer cells of a woman named Henrietta Lacks. More than 65,000 scientific studies using HeLa cells have been published since the 1950s, and the cells have been used to study every conceivable aspect of cell physiology as well as the basic machinery common to all cells (*Nature*, 2011, 480, 34-34). In addition, many researchers have used HeLa cells as disease models in photodynamic therapy (*Inorg. Chem.* 2021, 60, 16178–16193; *J. Am. Chem. Soc.* 2004, 126, 10619–10631; *Angew.Chem.Int.Ed.*2016,55,9947–9951). We use HeLa cells as a disease model for the following reasons: 1. HeLa cells are easy to culture and maintain. 2. More importantly, HeLa cells belong to the epithelial cells of cervical cancer (*Photoch Photobio Sci.* 2021 20, 1599–1609; *Front. Immunol.* 12:738431; *BMC Microbiology.* 2009, 9:139). For early phases of cervical carcinoma, the cryotherapy is very reasonable, however in case of late stage of the disease laser methods are used alone or on combination (*Gomal J Med Sci.* 2019, 17, 150-154), which means that cervical carcinoma is close to the cervical surface, thus intravenous injection may not be required during PDT. Therefore, the selection of HeLa cell model in this work may be feasible. However, we administered the photosensitizer by intratumoral injection, which may be more suitable for the treatment of skin cancer as mentioned by the reviewer. In the future work, we will try to use skin cancer as a PDT treatment model, such as melanoma.

Q: (11) In vivo study, there need a group of photosensitizers without GA targeting to support the advantage of GA targeting photosensitizer.

A: We thank the reviewer for this important comment and suggestion. To address this problem, we have performed additional experiments to compare the treatment effect of photodynamic therapy using two types of photosensitizers, one is the TPE-PyT-CPS and TPE-T-CPS, which are GA-targeting, and another photosensitizer is TPE-PyT-PS, which isn't GA-targeting. In the revised manuscript, we used ABDA as the singlet oxygen indicator to study the generation ability of singlet oxygen after laser irradiation of different photosensitizers (**Figure R2**). After irradiation, the decomposition rates of ABDA were 26.97, 27.91 and 32.85 $\mu\text{M}/\text{min}$ in TPE-PyT-PS, TPE-T-CPS and TPE-PyT-CPS (**Figure R2**), respectively. Interestingly, we found that TPE-T-CPS and TPE-PyT-PS have comparable singlet oxygen production, while TPE-T-CPS has golgi targeting capability but TPE-PyT-PS does not, which may be the key to explaining the advantages of golgi targeting therapy. The concentrated photosensitizers of 10 mM in DMSO were taken out of the refrigerator from 4°C and a

certain volume was taken out, and then diluted to 0.2 mM with normal physiological saline. The diluted photosensitizer was injected into mice in each administration group by intratumoral injection, in which the volume of mice tumor at the time of treatment was about $119.05 \pm 5.80 \text{ mm}^3$. In addition, then the mice in each experimental group were given the photosensitizer once every two days. Irradiation or not was selected at 18 h after photosensitizers injection according to group conditions, and tumor volume and body weight changes of mice were recorded every two days.

As can be seen from **Figure R18A**, for the control groups (“saline + light”, TPE-PyT-PS, TPE-T-CPS and TPE-PyT-CPS), the tumor volumes increased rapidly, suggesting that the the AIEgens has almost no inhibitory effect on tumor growth without light excitation. However, when the photosensitizer was excited by 532 nm laser, the tumor volume of mice in each light group (**Figure R18A, B**) was effectively inhibited to varying degrees compared with the control group. Specifically, TPE-T-CPS with Golgi targeting ability had better therapeutic effect ($p < 0.05$) than TPE-PyT-PS without Golgi targeting ability both in terms of tumor volume (**Figure R18B**) and tumor weight (**Figure R18C**) after PDT, indicating that Golgi targeting photosensitizers do have a more significant inhibitory effect on tumor growth. On the other hand, both of TPE-PyT-CPS and TPE-T-CPS can effectively target golgi apparatus, but TPE-PyT-CPS is significantly better than TPE-T-CPS in tumor growth inhibition ($p < 0.01$), indicating that pyrene introduced this molecular system to achieve effective charge separation within TPE-PyT-CPS, thus improving singlet oxygen generation ability, and realizing better tumor treatment effect.

Figure R18. Therapeutic effect of different AIEgens *in vivo*. (A) Photographs of excised tumors

on the 15th day in different groups (1. saline + L, 2.TPE-PyT-PS, 3.TPE-T-CPS, 4.TPE-PyT-CPS, 5.TPE-PyT-PS + L, 6.TPE-T-CPS + L, 7.TPE-PyT-CPS + L). (B) Changes of tumor volume and weight (C) during PDT that treated with AIEgens (0.2 mM, 100 μ L in saline) as the PS and 532 laser irradiations of 35 mW cm^{-2} for 5 min at different time points post-treatment in different groups (n =5 biologically independent animals, *p< 0.05, **p< 0.01). Data were presented as mean \pm SD. (D) Body weight curves of mice during PDT treatment in different groups (n =5 biologically independent animals). Data were presented as mean \pm SD. All data were presented as mean \pm SD. Statistical differences were analyzed by Student's t test. Source data are provided as a Source Data file.

Q: (11) The serum biochemistry index should be test to evaluate the safety of photosensitizer in vivo.

A: We thank the reviewer for this important comment and constructive suggestion. The in vivo toxicology and potential side effects were investigated systematically. Blood biochemical analysis were carried out and various parameters including aspartate transaminase (AST), alanine transaminase (ALT), blood urea nitrogen (BUN), creatinine (CREA), creatinine and alkaline phosphatase (GGT) were examined (**Figure. R19A**). Compared with the saline group, no meaningful (p value > 0.05) difference was detected from the five treated groups. Hence, the treatment did not affect the blood chemistry of mice. Furthermore, since alanine transaminase (ALT), aspartate transaminase (AST) and creatinine (CREA) are closely related to the functions of the liver and kidney of mice, the results demonstrated that the treatment induced no obvious hepatic and kidney toxicity in mice.

On the other hand, the standard haematology markers including the white blood cells (WBC), red blood cells (RBC), mean corpuscular volume (MCV), haemoglobin (HGB), mean corpuscular haemoglobin (MCH) and platelets (PLT) were measured (**Figure. R19B**). Compared with the saline group, all the parameters in the six treated groups appeared to be normal and the differences between them were not statistically significant (p value > 0.05). These results indicated that these treatments did not cause obvious infection and inflammation in the treated mice.

Figure R19. Investigation of the biosafety of AIEgens in mice after PDT. (A) Biochemistry and (B) Blood count analysis of mice after photodynamic therapy in different treating group (1. saline + L, 2.TPE-PyT-PS, 3.TPE-T-CPS, 4.TPE-PyT-CPS, 5.TPE-PyT-PS + L, 6.TPE-T-CPS + L, 7.TPE-PyT-CPS + L). Abbreviations: AST, glutamic oxaloacetic transaminase; ALT, alanine aminotransferase; BUN, blood urea nitrogen; GGT, γ -glutamyl transpeptidase; CRE, creatinine; RBC, red blood cells; WBC, white blood cells; HGB, hemoglobin; MCH, mean corpuscular hemoglobin concentration; PLT, platelet count; MCV, mean corpuscular volume. Values are mean \pm S.D. (n = 5). Source data are provided as a Source Data file.

Q: (12) The scale bar in Figure 5D is not very clear, please upload the picture again.

A: We sincerely thank the reviewer for your careful reading and point it out. In the revised manuscript, we have uploaded clear pictures of **Figure 5D** again.

Q: (13) There is no statistical analysis in any graphs. It would be essential in some of

them, such as Figure 10 C and 10 D for changes of tumor volume and weight after PDT.

A: We sincerely thank the reviewer for your careful reading and point it out. In the revised manuscript, we have made statistical analysis in the necessary pictures, including **Figure 10 C and D**.

Q: (14) Test are not distinct in Figure 10, Figure S19, Figure S20, Figure S25 and Figure S26. Please reupload carefully prepared pictures.

A: We sincerely thank the reviewer for your careful reading and point it out. In the revised manuscript, we have uploaded clear pictures again, including **Figure 10, Figure S19 (Figure S22 in revised manuscript), Figure S20 (Figure S28 in revised manuscript), Figure S25 (Figure S35 in revised manuscript) and Figure S26 (Figure S38 in revised manuscript)**.

Q: (15) In vivo PDT method is unclear. Please add the solvent of TPE-PyT-CPS and the tumor size when treatment was implemented.

A: We thank the reviewer for this important comment. In the PDT process, the concentrated photosensitizers of 10 mM in DMSO were taken out of the refrigerator from 4°C and a certain volume was taken out, and then diluted to 0.2 mM with normal physiological saline. The diluted photosensitizer was then injected into mice (100 µL in saline) in each administration group by intratumoral injection, in which the volume of mice tumor at the time of treatment was about $119.05 \pm 5.80 \text{ mm}^3$. In addition, then the mice in each experimental group were given the photosensitizer once every two days. Irradiation was selected at 18 h after photosensitizers injection, and tumor volume and body weight changes of mice were recorded every two days.

Reviewers' Comments:

Reviewer #1:

Remarks to the Author:

The authors have adequately revised their manuscript according to my previous comments and suggestions. The quality of the manuscript has been improved after the revision. I do not have further criticism of the work.

Reviewer #2:

Remarks to the Author:

The authors have made sufficient modifications according to the modification comments, and I suggest that this paper could be accepted without further modification.

Response to reviewers

Reviewer #1 (Remarks to the Author):

The authors have adequately revised their manuscript according to my previous comments and suggestions. The quality of the manuscript has been improved after the revision. I do not have further criticism of the work.

A: We thank this reviewer for a comprehensive and insightful review.

Reviewer #2 (Remarks to the Author):

The authors have made sufficient modifications according to the modification comments, and I suggest that this paper could be accepted without further modification.

A: We thank this reviewer for a comprehensive and insightful review.